# Reinforcement learning with learned gadgets to tackle hard quantum problems on real hardware

## Abstract

Designing quantum circuits for specific computational tasks remains a fundamental challenge in quantum computing, because of the exponential growth of the state space with the number of qubits. We propose *gadget reinforcement learning* (GRL), a framework that integrates reinforcement learning (RL) with program synthesis by automatically synthesizing composite gates, or "gadgets", and incorporating them into the RL agent's action space. This enables a more efficient exploration of the design space for parameterized quantum circuits (PQCs) that solve complex quantum tasks, such as approximating ground states of quantum Hamiltonians—an *NP-hard* problem.

We test GRL using the transverse field Ising model (TFIM), a standard testbed for quantum algorithms, under fixed computational budgets typical of research settings (e.g., 2–3 days of GPU runtime). Our experimental results demonstrate the advantages of GRL over baseline RL methods, including: (1) *Improved accuracy*: GRL achieves ground-state energy estimation up to machine accuracy; (2) *Hardware compatibility*: GRL generates compact PQCs that are more suitable for implementation on real quantum hardware, minimizing noise and gate errors; (3) *Scalability*: GRL exhibits robust performance as the size and complexity of the problem increases, even with constrained computational resources.

By integrating program synthesis into the RL framework, GRL facilitates the automatic discovery of reusable circuit components, specifically tuned for a given hardware. This bridges the gap between algorithmic design and practical quantum implementation. This makes GRL a versatile and resource-efficient framework for optimizing quantum circuits, with potential applications in hardware-specific optimizations, variational quantum algorithms, and other challenging quantum tasks.

## 1. Introduction

Quantum computing has experienced substantial advancements in recent years, unlocking the potential to solve classically intractable problems. Foundational algorithms like Shor's algorithm for integer factorization (Shor, 1999) and Grover's algorithm for unstructured search (Grover, 1996) demonstrate the transformative promise of quantum technology. However, practical implementation of these algorithms faces substantial hurdles due to the limitations of current quantum hardware, characterized by small qubit counts, significant noise, and constrained connectivity (Monz et al., 2016; Mandviwalla et al., 2018). These challenges require innovative approaches to bridge the gap between theoretical breakthroughs and hardware capabilities.

Hybrid quantum-classical algorithms, particularly variational quantum algorithms (VQAs), have emerged as a promising solution to this challenge. VQAs operate by dividing computation between quantum hardware and classical optimization. Their implementation involves three main steps: (1) Quantum state preparation: A parameterized quantum circuit (PQC) $U(\vec{\theta})$, containing adjustable parameters $\vec{\theta}$, is constructed using single-qubit rotations and non-parameterized two-qubit entangling gates. (2) Measurement: The PQC is executed on quantum hardware to evaluate the cost function:

$$C(\vec{\theta}) = \langle 0|U^{\dagger}(\vec{\theta})HU(\vec{\theta})|0\rangle, \qquad (1)$$

where $H$ represents the Hamiltonian encoding the problem. (3) Optimization: Classical algorithms minimize $C(\vec{\theta})$ by adjusting $\vec{\theta}$. This paradigm transforms the challenge of solving a quantum problem into finding an optimal PQC that minimizes the cost function.

However, designing effective PQCs remains difficult due to the constraints of current quantum hardware. Different noise levels, qubit connectivity topologies, and gate fidelities complicate the process, making hardware-specific

---

[1]Anonymous Institution, Anonymous City, Anonymous Region, Anonymous Country. Correspondence to: Anonymous Author <anon.email@domain.com>.

Preliminary work. Under review by the International Conference on Machine Learning (ICML). Do not distribute.

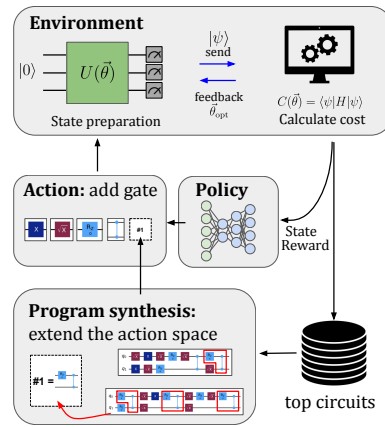

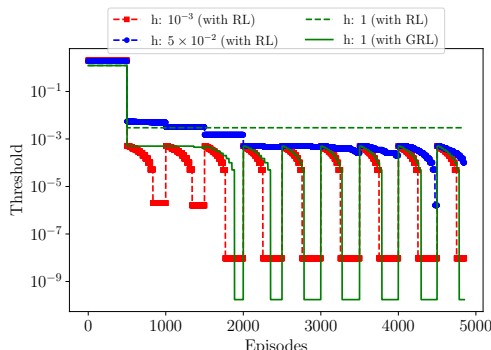

*Figure 1.* (left) **Sketch of the gadget reinforcement learning algorithm**. A reinforcement learning agent sequentially adds gates to a circuit for the preparation of a quantum state. The expectation value of the energy of a given Hamiltonian is calculated and used as cost. The parameters $\vec{\theta}$ of the constructed circuit are optimized to minimize the cost. A reward is provided to the reinforcement learning agent according to a threshold $\zeta$: if the cost is smaller, the agent gets a reward $r$, otherwise $-r$. Subsequently, the reward is used to improve the policy. The algorithm stores the top $k$ circuits. After a training loop, those circuits are analyzed with the program synthesis algorithm and gadgets, i.e. composite gates that are most likely to be useful, are proposed. The best ones are added to the available actions, extending the action space. The reinforcement learning agent will then start a new training loop. (right) **Comparison of the best solutions to our example application in different regimes.** We consider the problem of finding the ground state of a 2-qubit transverse field Ising model, as defined in Eq. 4. Different regimes are given by varying the magnetic field strength $h$, with $h = 10^{-3}$ being the simplest task, and $h = 1$ the most challenging task. A pure reinforcement learning agent significantly declines in performance as the difficulty of the problem increases. On the other hand, gadget reinforcement learning can solve also the hardest regime, $h = 1$, which cannot be solved with the RL approach.

PQC design particularly challenging. Recent efforts have focused on adaptive methods (Grimsley et al., 2019; Tang et al., 2021; Feniou et al., 2023) and advanced optimization techniques (Zhou et al., 2020; Zhu et al., 2022; Cheng et al., 2024; Kundu et al., 2024a) to address these issues. Additionally, machine learning approaches, particularly reinforcement learning (RL), have emerged as promising tools for automating PQC design (Krenn et al., 2023; Bang et al., 2014; Ostaszewski et al., 2021; Kundu, 2024).

In conventional RL-based approaches, an agent explores a fixed action space comprising predefined quantum gates to construct PQCs. While effective for certain problems, this approach limits the agent's adaptability and scalability. To overcome these limitations, curriculum RL (CRL) has been introduced, allowing the agent to progress through tasks of increasing complexity (Patel et al., 2024b). Despite these advancements, both fixed-action-space RL and CRL are limited by computational resources. With a fixed action space, agents often require extensive exploration to identify optimal solutions, which results in suboptimal utilization of computational resources. This inefficiency becomes more apparent in complex tasks, where managing the computational budget becomes crucial[1].

Addressing this challenge requires a framework capable of leveraging insights from simpler problems to solve more complex ones efficiently, within a fixed computational budget. This paper introduces *gadget reinforcement learning* (GRL), a novel approach that combines RL with program synthesis (PS) to dynamically expand the agent's action space. GRL achieves this by synthesizing higher-level composite gates, or "gadgets", from the solutions of simpler problem instances and incorporating these gadgets into the agent's action space. By doing so, GRL enhances the agent's ability to generalize and adapt, optimizing computational resource utilization.

A schematic of the GRL algorithm is presented in Fig. 1 (left). The process begins with an RL agent solving a simple instance of a problem using a basic action space, such as the native gateset of a specific quantum processor. The program synthesis component identifies recurring patterns in the top-performing circuits, synthesizes these patterns into gadgets, and adds them to the action space. With this expanded action space, the RL agent retrains to tackle more challenging problem instances.

To demonstrate the efficacy of GRL, we apply it to the transverse field Ising model (TFIM), a problem that becomes increasingly difficult as the magnetic field strength $h$ or the system size grows. GRL learns gadgets from a simple 2-

---

[1]Due to the limitations of the available cluster, we restricted the training to 5000 episodes and a maximum of 48 hours of runtime.

qubit TFIM with $h = 10^{-3}$ and successfully uses them to solve more complex instances, including a 3-qubit TFIM at $h = 1$, a regime where conventional RL approaches fail due to computational limitations. The comparison of performance across regimes is shown in Fig. 1 (right).

Our results highlight the advantages of GRL: (1) *Improved computational efficiency*: By learning and leveraging gadgets, GRL achieves superior performance within a fixed computational budget, avoiding the exhaustive exploration required in fixed-action-space RL. (2) *Scalability*: GRL effectively generalizes knowledge from simpler tasks to more complex ones, reducing the computational burden associated with solving larger problems. (3) *Hardware compatibility*: The PQCs generated by GRL are compact and hardware-optimized, making them more resilient under noise and practical for real-world implementation.

Specifically, GRL achieves up to a $10^7$-fold improvement in error reduction for ground-state energy estimation compared to baseline RL approaches under a fixed computational budget. Moreover, by using gadgets learned from simple TFIM instances, GRL can solve larger systems with fewer parameters and gates, demonstrating its efficiency and scalability.

The remainder of this paper is organized as follows: Sec. 2 reviews prior work on RL and PS in quantum systems; Sec. 3 details the GRL algorithm, and Sec. 4 benchmarks GRL and shows its advantage over state-of-art RL in solving TFIM in simulated environment and quantum hardware. Finally, Sec. 5 discusses the implications of our findings and outlines future research directions.

## 2. Related works

**Reinforcement learning for quantum computing** Reinforcement learning (RL) has become one of the most effective techniques for optimizing parameterized quantum circuits (PQCs) in variational quantum algorithms. These methods typically rely on carefully designed reward functions to train agents for selecting suitable quantum gates. In (Ostaszewski et al., 2021), double-deep Q-Network (DDQN) combined with an $\epsilon$-greedy policy was employed to estimate the ground state of chemical Hamiltonians. Similarly, (Ye & Chen, 2021) proposed a DQN-based framework with an actor-critic policy and proximal policy optimization to construct multi-qubit maximally entangled states.

In (Fösel et al., 2021), a novel deep reinforcement learning approach was introduced for quantum circuit optimization, demonstrating improved circuit efficiency and supporting hardware-aware strategies. Expanding on this, (Patel et al., 2024b) applied curriculum reinforcement learning and advanced pruning techniques to address PQC challenges on realistic quantum hardware. The work in (Tang et al., 2024) integrated reinforcement learning with Monte Carlo tree search to minimize routing overhead in quantum circuits.

Further, (Foderà et al., 2024) demonstrated how RL could autonomously generate quantum circuits as ansatzes in variational quantum algorithms (VQAs), solving diverse problems and introducing novel ansatz families, such as for the Maximum Cut problem. Reinforcement learning methods employing cost explosion strategies to enhance training efficiency and reach optimal quantum circuits were proposed in (Moflic & Paler, 2023). In (Kundu, 2024), a novel encoding of PQCs, combined with a dense reward function and $\epsilon$-greedy policy, tackled the quantum state diagonalization problem. Additionally, RL was shown to address hard instances of combinatorial optimization problems, surpassing the performance of state-of-the-art algorithms, as evidenced in (Patel et al., 2024a). Insights from quantum information theory were leveraged in (Sadhu et al., 2024) to guide RL agents in prioritizing architectural features for improved PQC search and optimization.

Despite these advances, a significant limitation of existing RL approaches lies in their fixed action space, which can lead to performance degradation as the number of qubits or problem complexity increases. Moreover, many of these methods rely on non-native gates that require transpilation to fit specific hardware constraints, necessitating additional noise-resilience techniques.

**Program synthesis in quantum domain** Program synthesis has shown promise for expanding the action space of reinforcement learning agents, enabling the discovery of high-level actions, often referred to as option discovery or skill learning in RL literature. Techniques explored in this domain include learning multiple policies with a meta-controller, leveraging information theory, and maximizing diversity (Bacon et al., 2017; Nachum et al., 2018; Machado et al., 2024; Krishnan et al., 2017; Frans et al., 2018; Gregor et al., 2016; Florensa et al., 2017; Eysenbach et al., 2019).

Inspired by recent advancements in program synthesis, such as DreamCoder (Ellis et al., 2020), simpler task solutions can be analyzed to extract common fragments as new primitives, thereby reducing search complexity. Even naive enumeration approaches benefit significantly from this compression technique (Dechter et al., 2013). For quantum computing, program synthesis has been successfully applied to decompose unitary matrices (Sarra et al., 2024). This work generalizes the synthesis process by replacing brute-force enumeration with a reinforcement learning agent for the search step.

Additionally, composite gates, known as gadgets, have been introduced to simplify quantum circuit searches (Ruiz et al., 2024). While pattern recognition methods have been used to extract gadgets for interpretability (Trenkwalder et al., 2023), their iterative application to enhance the performance

of RL agents remains underexplored.

## 3. Methods

We propose a general technique to build circuits that solve quantum optimization problems. Our approach combines a reinforcement learning agent to search for the parameter-ized quantum circuit (PQC) space with a program synthesis algorithm that analyzes the best circuit to extract gadgets (i.e., new composite components) that extend the agent's action space. This method, which we call gadget reinforce-ment learning (GRL), is particularly useful when solving a parametrized class of problems. Especially when the problem has different degrees of difficulty according to its parameters, we can learn a set of operations from simpler problems and use them subsequently to help solve the harder ones.

### 3.1. Gadget reinforcement learning

We provide an overview of the GRL algorithm for construct-ing PQCs in a VQA task. Consecutively, we provide details on the state and action representations as well as the reward function employed in this study.

The GRL algorithm initiates with an empty quantum cir-cuit. The RL agent, based on a double deep Q-network and $\epsilon$-greedy policy (for further details see Appendix D.2), sequentially appends the gates to the circuit until the max-imum number of actions has been reached. The actions are chosen from an action space of available elementary gates. In particular, in our application, it contains RZ, SX, X as single qubit gates, where RZ is the only parameter-ized gate in the action space. Furthermore, to entangle the qubits we use Controlled-Z (CZ) gate. The main motive to choose such an action space is that all these gates are na-tive gateset of the newly introduced IBM Heron processor. Therefore, we do not need to further transpile the circuits, which is an NP-hard task (IBM Quantum Documentation, 2024), when executing on the processor, apart from remov-ing possible gate sequences that simplify to identity. We implement a double deep RL method, where the PQCs are encoded in a refined binary tensor representation, as intro-duced in (Kundu et al., 2024b). This encoding is inspired by the tensor-based encoding introduced in (Patel et al., 2024b). In the Appendix D.1 we elaborately describe the refined encoding scheme with an example.

To steer the agent towards the target, we use the same re-ward function $R$ at every time step $t$ of an episode, as in (Os-taszewski et al., 2021). The reward function $R$ defined as,

$$R = \begin{cases} r, & \text{if } C_t < \zeta, \\ -r & \text{if } t \geq T_{\max} \text{ and } C_t \geq \zeta, \\ \mathcal{C}, & \text{otherwise.} \end{cases} \quad (2)$$

where $\mathcal{C} = \max\left(\frac{C_{t-1}-C_t}{|C_{t-1}-C_{\min}|}, -1, 1\right)$, $r$ is a real positive number, $C_t$ represent the value of the cost function $C$ (as defined in Eq. 1) at step $t$, and $T_{\max}$ denote the maximum number of steps allowed for an episode. Additionally, note that when the agent receives a positive reward value $r$, the episode concludes. In other words, there are two stopping conditions: either surpassing the threshold $\zeta$ or reaching the maximum number of actions. The agent's objective is to estimate the value of $C_{\min}$ with the desired precision $\zeta$.

In what follows, we utilize a feedback-driven curriculum re-inforcement learning agent. In particular, the agent updates its threshold while running the episodes: if we find a ground state with lower energy than the threshold, we decrease the threshold, otherwise, we increase it again. The algorithm is described with more technical detail in Appendix B.

In the next step, we sample the top $k$ PQCs, chosen accord-ing to how effective they are at estimating the solution to the problems, i.e., with smaller associated $\zeta$ value. These PQCs are then processed through a program synthesis (PS) algorithm, as described in Section 3.2. By considering an appropriate tradeoff between the proposed component usage frequency and its complexity (simpler components are more likely to generalize), we can extract composite gates, i.e., *gadgets*, by choosing those with the largest log-likelihood. We *gadgetize* the RL algorithm by updating the action space with the *gadgets* discovered by the library building module. Finally, the GRL is executed again with the modified action space, consisting of the initial gateset corresponding to the quantum hardware and the *gadgets*.

### 3.2. Library building

To update the action space in GRL, a library-building algo-rithm that leverages a program synthesis framework inspired by (Sarra et al., 2024; Ellis et al., 2020) is employed. The algorithm analyzes the top-$k$ PQCs to identify and extract common, useful gate sequences and structures. The PQCs are expressed as programs in a typed-$\lambda$-calculus formal-ism (Pierce, 2002), where the gates act as functions that take a quantum circuit and the target qubits as inputs and return the updated PQC with the gate applied. For example, a function that applies an $X$ gate on the first qubit and then a controlled-$Z$ gate can be represented as

$$f(I_2) = cz(x(I_2, 0), 0, 1) \quad (3)$$

where $I_2$ is a 2-qubit empty circuit. Each circuit program is organized into a syntax tree. The algorithm decomposes each circuit into fragments, i.e. sets of operations, and looks for the most common fragments in the input set. We use the fragment grammar formalism to evaluate each fragment's usefulness based on a grammar score. In this context, a grammar $g$ consists of elementary gates (primitives) with usage probabilities estimated from the given set of $k$ top

circuits. The grammar score function prioritizes grammars that are most likely to produce effectively the given set of circuits, while balancing complexity.

We then modify the action space of the RL agent by adding the highest scoring fragments, which are expected to help find more compact PQCs with a smaller number of gates. In our experiments, we show that, although the library is built upon problems that are small and simple, these libraries generalize effectively and can be utilized to *gadgetize* RL and solve harder instances of the given problem iteratively. For further details on grammar scoring, fragment grammar structure, and hyperparameter settings, refer to Appendix C.

The GRL runs iteratively by first considering a small system, (e.g. in our case a 2-qubit Ising model in a weak transverse field, $h = 10^{-3}$) and finding the solution within a predefined threshold ($\zeta$). The agent then finds the ground state within the compute budget, expressed by a fixed number of episodes. Subsequently, we try to solve an intermediately difficult problem (in our case, the Ising model with a larger transverse field, $h = 5 \times 10^{-2}$).

## 4. Results

As an example application for our algorithm, we consider the transverse field Ising model (TFIM). The goal is to design a circuit which finds the ground state of the system, i.e. the system with the lowest energy. This problem is well-known to be NP-Hard (O'Connor et al., 2022). The system is defined by

$$H = -J \sum_{\langle i,j \rangle}^{N} \sigma_i^z \sigma_j^z - h \sum_i \sigma_i^x \qquad (4)$$

where $N$ is the number of qubits, $J$ is the coupling constant between neighboring spins, $h$ is the strength of the transverse field, $\sigma_i^z$ and $\sigma_i^x$ are the Pauli matrices acting on the $i$-th spin in the $z$- and $x$-direction, respectively, and $\langle i, j \rangle$ denotes summation over nearest neighbors. This model presents a ferromagnetic phase transition at $J \gg h$ and has been studied thoroughly in the literature, for example, with hybrid quantum-classical approaches (Sumeet et al., 2023) where they utilize numerical linked-cluster expansions with the variational quantum eigensolver (VQE) for TFIM with one-dimensional chains and the two-dimensional square lattice.

The primary motivation for using the transverse field Ising model (TFIM) in this problem is the increasing difficulty in finding the ground state as the magnetic field strength $h$ varies from small values (on the order of $10^{-3}$) towards 1, which is defined as the phase change point. This difficulty arises due to the degeneracy between the ground and first excited states that emerge as $h$ approaches the critical

value (Curro et al., 2024; Pfeuty, 1970). As shown in Appendix A the degeneracy phenomenon is a key feature of the quantum phase transition in the TFIM and significantly impacts the behavior of the system near the critical point.

The primary objective of employing gadget reinforcement learning (GRL) is to derive *gadgets* from easily solvable instances (in our case, where $h \ll J$) through program synthesis within an RL framework. These gadgets are then utilized to modify the action space in the RL framework, enabling efficient solutions for more challenging instances. We quantify this efficiency through two key metrics: (1) Agent performance: This is evaluated by analyzing the cumulative reward, the nature of agent-environment interactions, and the total training duration. (2) Training accuracy: We assess how accurately the GRL agent performs in comparison to state-of-the-art RL agents. Our focus is on quantifying the number of 1- and 2-qubit gates required to achieve a specified accuracy, both in simulated environments and on actual quantum hardware.

### 4.1. Improved performance

We recall that GRL runs iteratively, with the agent and environment specifications as provided in Appendix D.3. Additionally, in Appendix H we provide an elaboration of the training time by both the RL- and GRL-agents.

**Agent accuracy and success frequency**  Fig. 2 summarizes the performance of RL and GRL agents in finding the TFIM ground state. The RL-only framework starts with a small system in an easy regime (e.g., weak transverse field, $h = 10^{-3}$) and achieves machine precision within a fixed compute budget (up to 48 hours). For the intermediate regime ($h = 5 \times 10^{-2}$), the agent finds an approximation, but the PQCs are large, and the errors are relatively high compared to their size. Notably, the RL-agent fails to give us a good approximation of the ground state for $h = 1$ and the number of successful episodes[2] drastically reduces as we increase the precision.

To improve efficiency, we analyze the top $k$ PQCs from earlier cases and extract key components as new primitive composite gates, or *gadgets*. By adding the most likely gadget to the RL agent's action space, we achieve significantly better approximations of the ground state. As additional gadgets are included, the agent experiences increasingly frequent successful episodes, achieving progressively lower error in estimating the ground state.

Compared to a state-of-the-art curriculum-based RL approach (Patel et al., 2024b), the gadget-based GRL agent is more effective, particularly in harder regimes. Gadget

---

[2]An episode is deemed successful if the agent approximates the ground state within a predefined threshold $\zeta$.

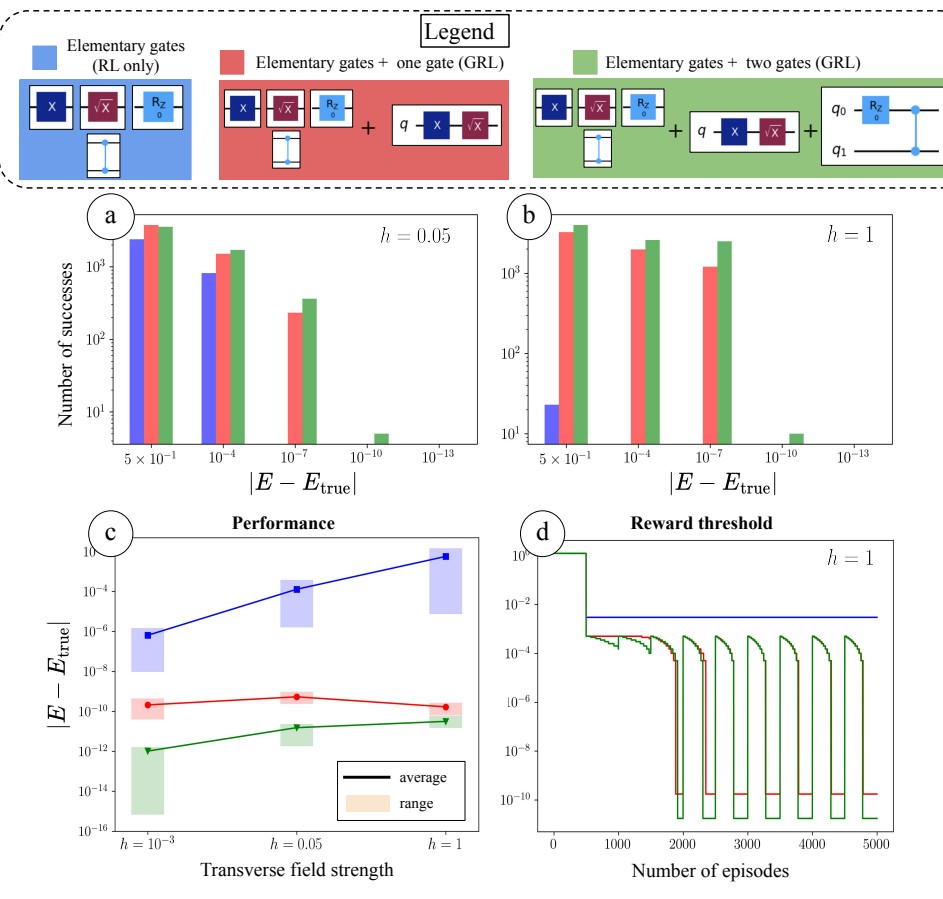

*Figure 2.* **Results for the 2-qubit transverse field Ising model (TFIM)**. We compare reinforcement learning-only (blue) with gadget reinforcement learning (GRL) using one (reddish orange) and two (green) extracted components, as shown in the legend. (a) and (b) show the number of successful episodes (agent finds the ground state within predefined accuracy) for $h = 5 \times 10^{-2}$ and $h = 1$ (phase change point), respectively. (c) compares error scaling with varying transverse field strength under a fixed compute budget (48-hour GPU run). Solid lines show averages over multiple runs; shaded areas indicate solution ranges (smallest values are most relevant). GRL achieves high accuracy for $h = 1$. (d) plots RL reward thresholds during training for $h = 1$, showing GRL finds circuits with lower cost. Without gadget extraction, accuracy is limited to $10^{-3}$, while GRL achieves machine precision.

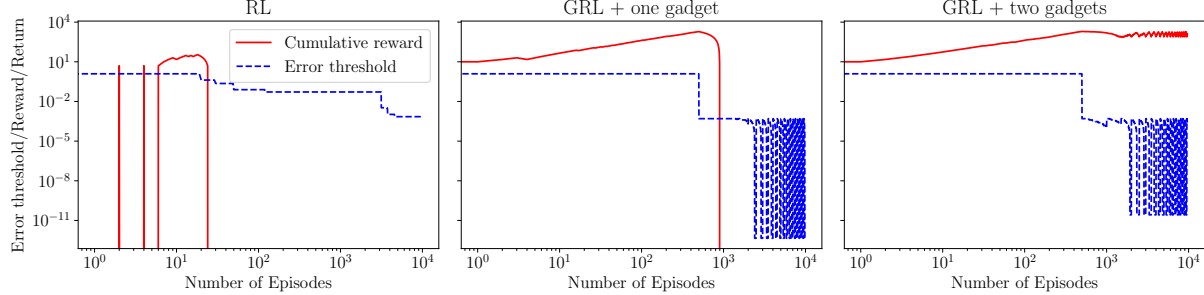

*Figure 3.* **For $N = 2$ TFIM, GRL with one and two gadgets improves the cumulative reward growth compared to RL**. The RL-agent struggles with consistent cumulative rewards and positive returns. GRL with one gadget improves performance, achieving steady cumulative reward growth and frequent positive returns but experiences a notable drop around the 1000th episode, reaching the machine precision error threshold. GRL with two gadgets resolves this drop and further enhances performance.

extraction is performed on easier tasks, and the resulting          modified action space is used to tackle more challenging

| Problem | Method | Metric | #CZ | #RZ | #SX | #X |
|---------|--------|--------|-----|-----|-----|-----|
| 2-qubit TFIM | GRL | Average | 2.0 | 6.0 | 4.43 | 2.0 |
| | GRL | Minimum | 2 | 5 | 4 | 1 |
| | RL | Average | 2.0 | 8.08 | 6.62 | 1.75 |
| | RL | Minimum | 1 | 6 | 4 | 1 |
| 3-qubit TFIM | GRL | Average | 6.0 | 11.0 | 11.0 | 1.0 |
| | GRL | Minimum | 2 | 9 | 7 | 1 |
| | RL | Average | 8.83 | 21.67 | 22.67 | 1.0 |
| | RL | Minimum | 7 | 27 | 27 | 1 |

*Table 1.* **Length and composition of constructed circuits.** We compare gadget reinforcement learning (GRL) with state-of-the-art curriculum reinforcement learning in the hardest regime ($h = 1$). Results are based on transpiling top-performing circuits for the `IBM Heron` processor. GRL achieves smaller gate counts compared to RL-only. The averages reflect PQCs with similar errors, while the minimum metric represents the most compact transpiled PQC with comparable performance.

problems, such as $h = 1$. This demonstrates the utility of the approach, especially for complex scenarios.

**Cumulative rewards and returns**  Fig. 3 shows that the RL-only agent struggles with consistent rewards and positive returns, while GRL with one gadget steadily improves but plateaus around the 1000th episode, reaching machine precision. Adding a second gadget resolves this drop and enhances performance. It is worth noting that for the $N = 2$ qubit problem, machine precision can be achieved with just one extracted gadget, making the second gadget redundant in this case. The two-gadget implementation for $N = 2$ serves to illustrate the potential for performance improvement when additional gadgets are introduced in more complex systems. A similar observation is recorded for $N = 3$ qubit TFIM and is in Appendix F.

**Generalization of extracted gadgets**  Fig. 4 demonstrates the performance of GRL agents using 2-qubit gadgets for $N = 3$. While one gadget achieves machine precision for $N = 2$, it only reaches an error of $10^{-4}$ for $N = 3$. Adding a second gadget dramatically improves performance, enabling machine precision in both easy and hard regimes. In contrast, the RL-only agent fails to learn due to the search depth required, especially in the hard regime ($h = 1$), where it produces high-error solutions likely assembled randomly. Additional gadgets allow RL to find significantly better solutions. For a comprehensive analysis of our ablation study and detailed numerical results, please refer to Appendix E.

### 4.2. Found circuits are suitable for real hardware

**More compact circuit for real hardware**  We compare PQCs from gadget reinforcement learning (GRL) with state-of-the-art RL methods for finding the TFIM ground state.

We benchmark against curriculum reinforcement learning (Patel et al., 2024b) using a universal gateset (RX, RY, RZ, CX) as in (Patel et al., 2024b; Kundu, 2024), comparing it to GRL with an extended action space including gadgets. The GRL action space incorporates the `IBM Heron` processor's native gateset and composite gates derived from top-performing PQCs for 2-qubit TFIM at $h = 10^{-3}$ and $h = 5 \times 10^{-2}$. We estimate the ground state of 2-qubit and 3-qubit TFIM at the phase change point ($h = 1$). GRL-obtained circuits achieve similar error to RL but the circuits are more compact when transpiled for real quantum hardware. Table 1 summarizes results after transpiling in `IBMQ Torino` (part of `IBM Heron` processor).

Moreover, Appendix I details hardware topology and in Appendix G we show GRL uses $3\times$ fewer CZ, RZ, and SX gates for similar error (in the order of $\sim 10^{-4}$) in 3-qubit TFIM. This suggests an advantage in solving problems directly with GRL consisting of target hardware components and gadgets, rather than first finding solutions in a universal gateset and then transpiling for the target hardware.

**Improved performance on real hardware**  In Appendix G we further show that the circuits obtained in the noiseless scenario by GRL provide better approximation to the ground state estimation for both the 2- and 3-qubit TFIM across multiple quantum hardware modules on the `IBM Heron` and `IBM Eagle` processors. We emphasize that no constraint on the circuit depth has been enforced in the GRL agent, even though this can be considered in future applications to encourage shorter circuits, or avoid using expensive gates.

## 5. Outlook

In this paper, we have shown how to learn reusable components from different regimes for efficiently building quantum circuits that solve some given problems. Instead of considering a single specific problem, we start from a trivial regime and gradually tackle the harder one. By finding the ground state in the low transverse field regime, we discover sequences of gates that are recurrent, and we can extract them as gadgets and use them to extend the action space of subsequent iterations. This proves to be very effective because it largely reduces the required depth of the circuit at the cost of a slightly increased breadth of the search. In other words, the extracted gates serve as a data-driven inductive bias for solving the given class of problems.

In terms of shortcomings of our approach, the main overhead to consider is the necessity of performing multiple iterations. In particular, it is important that the target class of problems has a structure with different degrees of difficulty: if the problem is too difficult, the reinforcement learning agent does not receive any signal, it will only learn

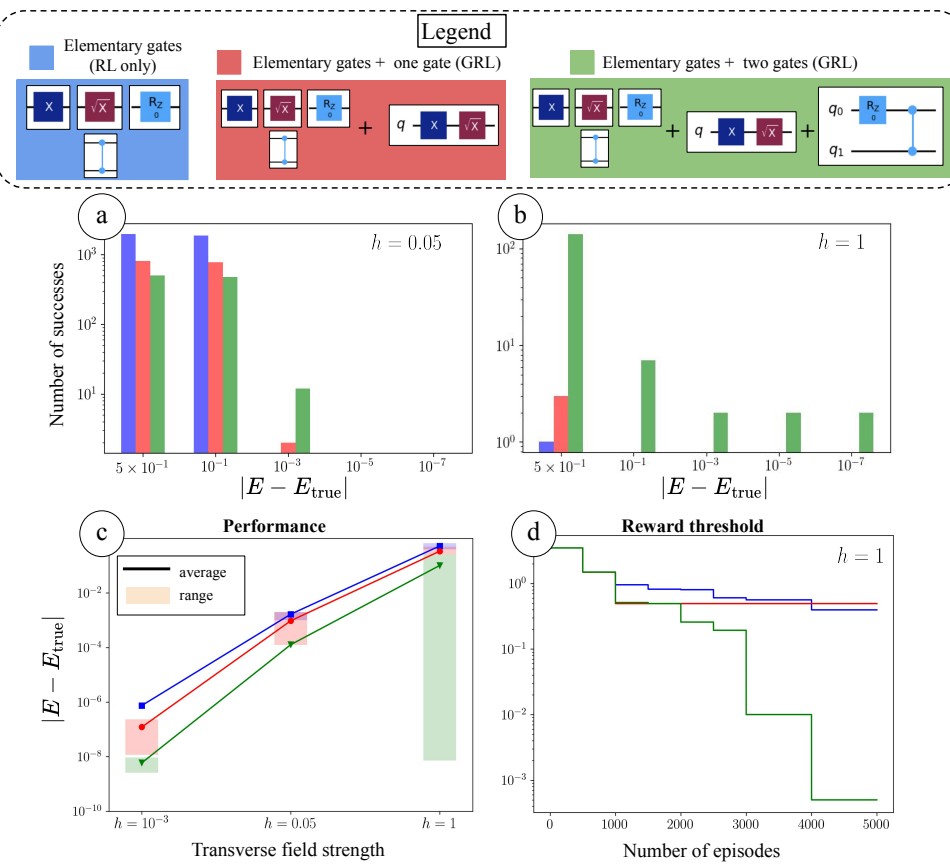

*Figure 4.* **GRL with two gadgets (green) overcomes the training bottleneck of RL-only (blue) and GRL with one gadget (red).** As in Fig. 2, subplots (a) and (b) show successful episodes, while (c) and (d) compare error scaling and RL reward thresholds. Under a fixed compute budget and varying transverse field strength, RL and GRL with one gadget achieve accuracies around $0.1$, whereas GRL with two gadgets attains machine precision. It should be noted that the gadgets that are used in this simulation are the same ones extracted while solving the $N = 2$ qubit TFIM.

to produce random circuits and the extracted gates will not be necessarily useful. On the other hand, if one regime is trivial and the other one is too hard, there is a low chance of generalization. Also, to extend the actions of the reinforcement learning agent multiple approaches are possible. In our example, we reinitialized the agent after extending the action space. However, smarter approaches, for example by just adding extra output neurons at the last layer of the policy, associating them to the added gadgets, may allow starting from the previous policy, while adding a small bias to encourage the exploration of the new action.

Our technique is general and can be extended to other quantum problems. For instance, we can efficiently solve challenging correlated quantum chemistry instances by leveraging gadgets from simpler ones. Easy instances involve smaller action spaces, and as the action space grows or accuracy requirements for the ground state increase, the problem becomes more difficult (McCaskey et al., 2019; de Gracia Triviño et al., 2023). This approach can also be

applied to quantum optimization, simulation, and machine learning, where easy instances help address more complex scenarios. Furthermore, it may be suitable for real hardware optimizations. Indeed, it allows to explicitly define the elementary gates to use for the decomposition, as opposed to finding the solution in a high-level gate set first (e.g. rotation gates RX, RY, and RZ) and transpiling them later. This can arguably produce more efficient circuits. Also, penalties for the length of the circuit or for the use of specific gates could be enforced, encouraging gates that are more reliable or cheap to implement on real hardware. In addition, the elementary components could also be modified to include some model of the noise on the real hardware, thus possibly finding a solution for some quantum problem that already includes some noise mitigation effects.

**Reproducibility**   The code used to generate the results are under preparation.

## 6. Impact statement

This work advances the field of quantum computing and machine learning by introducing gadget reinforcement learning (GRL) for efficient quantum circuit design. The potential broader impacts include:

- *Design hard problems by learning gadgets*: GRL could significantly speed up the design of computationally complex problems by learning gadgets from easy-to-solve ones, potentially leading to breakthroughs in various fields such as cryptography, drug discovery, and materials science.

- *Democratization of quantum computing*: The automated circuit design process could make quantum algorithm development more accessible to researchers without deep expertise in quantum physics, broadening participation in the field.

- *Energy efficiency*: GRL can tackle difficult quantum tasks in a fixed computation budget, providing more efficient quantum circuits. This may lead to reduced energy consumption in quantum computations, contributing to sustainable computing practices.

- *Ethics and safety*: This work enhances the capabilities of reinforcement learning agents, which are general-purpose algorithms that can be used for different purposes than quantum computing.

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

## A. The Transverse Field Ising Model: different regimes

As shown in Fig. 5, by looking at the ground state energy gap, we can identify three different regimes in the Transverse Field Ising Model:

1. in the low external field, the first excited state is almost degenerate with the ground state, therefore it is easy to find a low-energy state;

2. the regime where $h \simeq 0.1$, where the energy gap increases and the ground state starts to have a visibly different energy from the first excited state;

3. $h \gg 0.1$ where the energy gap is larger and the ground state energy is much smaller than that of the first excited state.

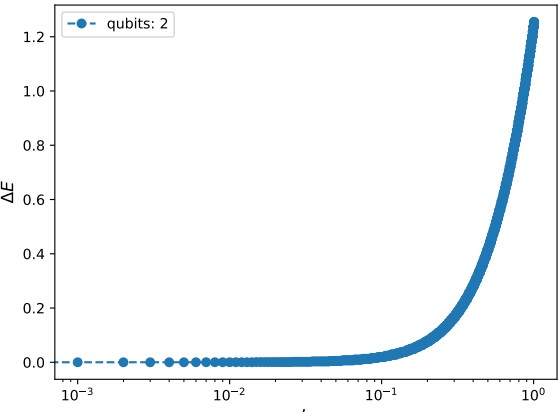

*Figure 5.* Energy gap between first excited state and ground state of the TFIM model as a function of the transverse field strength. The separation $\Delta E$ is negligible till $h = 10^{-1}$. Hence, due to energy degeneracy, it is easy to find a good energy approximation for $h \leq 10^{-1}$. The problem becomes harder when we choose $h \geq 10^{-1}$ as $\Delta E$ becomes non-negligible.

## B. Feedback-driven curriculum reinforcement learning

During learning, the agent maintains a pre-defined threshold $\zeta_2$ representing the lowest energy observed so far, updating it based on defined rules. Initially, $\zeta_2$ is set to a hyperparameter $\zeta_1$. When a lower threshold is found, $\zeta_2$ is updated to this new value. A *fake minimum energy* hyperparameter, $\mu$, serves as a target energy, approximated by the following:

$$\text{fake minimum energy} = (N-1) \times (-J) + N \times (-h), \tag{5}$$

where $N$ is the number of interacting spins, $J$ is the coupling strength between the spins and $h$ is the strength of the magnetic field.

Without amortization, the threshold updates to $|\mu - \zeta_2|$ when $\zeta_2$ changes; with amortization, it becomes $|\mu - \zeta_2| + \delta$, where $\delta$ is an amortization hyperparameter. The agent then explores subsequent actions and records successes.

Two threshold adjustment rules apply: a greedy shift to $|\mu - \zeta_2|$ after $G$ episodes (where $G$ is a hyperparameter) and a gradual decrease by $\delta/\kappa$ with each successful episode, where $\kappa$ is a shift radius hyperparameter. If repeated failures occur after setting the threshold to $|\mu - \zeta_2|$, it reverts to $|\mu - \zeta_2| + \delta$, allowing the agent to backtrack if stuck in a local minimum.

## C. Detailed description of the Library Building algorithm

The library building algorithm analyzes a set of circuits $\mathcal{D}$ in relation to a given set of elementary gates $g$, called grammar. In this framework, each grammar $g$ consists of elementary gates (also called "primitives") with assigned probabilities based

on usage frequency in the dataset. When we consider whether we should add a new gadget to our set of elementary gates, we compare the grammar with and without the new gadget, and accept the new gate if we improve the grammar score. This quantity evaluates how good a grammar is to represent the given dataset $\mathcal{D}$, trading off the likelihood of sampling circuits from the dataset with an approximation of the complexity of the grammar itself. In particular, given a set of circuits $\mathcal{D}$, we define the grammar score $S$, representing the grammar's efficiency in describing the circuits, as

$$S_{\mathcal{D}}(g) = L_g(\mathcal{D}) - \lambda|g| - k \sum_{p \in g} |p|, \tag{6}$$

where:

- $|g|$: the number of components in grammar $g$,

- $p$: a component or building block in the grammar,

- $|p|$: the number of elementary gates in $p$,

- $\lambda = 1$ and $k = 1$ are hyperparameters.

The first term represents the likelihood of reproducing the observed circuits, while the last two terms are complexity regularizers, inspired by the minimum description length principle. The likelihood $L_g(\mathcal{D})$ of a grammar is approximated with the probability of randomly sampling the circuits in the dataset using the grammar gate probabilities. Each circuit is weighted by the accuracy in solving the task (measured as the opposite of the energy).

The main hyperparameters that we consider to tune the algorithm are:

1. **Arity**: This controls the maximum number of arguments a component can have, or equivalently, the maximum number of qubits an extracted gate can act on. Here, we set arity $= 2$.

2. **Pseudocounts**: A constant shift in the usage frequency, which adjusts the log-likelihood estimation by ensuring each component is treated as though it is used at least once, even if unobserved. This allows patterns to be considered useful only if they appear frequently in the dataset. We set pseudocounts $= 10$.

3. **Structure Penalty** $k$: This regularizes the tradeoff between grammar likelihood and complexity. Lower penalties yield higher likelihoods but may overfit, while higher penalties result in simpler grammars that generalize better. We set structurePenalty $= 1$.

For a more technical descriptions of the $\lambda$-calculus tree structures and their efficiency, see (Ellis et al., 2020; Sarra et al., 2024).

## D. Implementation details

### D.1. Quantum circuit encoding

We employ a refined version of the tensor-based binary encoding introduced in (Kundu et al., 2024b), which is inspired by the encoding presented in (Patel et al., 2024b), to capture the architecture of a parametric quantum circuit (PQC), specifically by encoding the sequence and arrangement of quantum gates. Unlike the encoding presented in (Patel et al., 2024b), which is only the function of the number of qubits $N$, the refined encoding is a function of $N$ and the number of 1-qubit gates $N_{1q}$. This makes it suitable for the encoding of a broad range of action spaces and enables the agent to access a complete description of the circuit. To ensure a consistent input size across varying circuit depths, we construct the tensor for the maximum anticipated circuit depth.

To build this tensor, we define the hyperparameter $T_{max}$, which restricts the number of allowable gates (actions) across all episodes. A *moment* in a PQC refers to all simultaneously executable gates, corresponding to the circuit's depth. We represent PQCs as three-dimensional tensors where, at the start of each episode, an empty circuit of depth $T_{max}$ is initialized. This tensor is dimensioned as $[T_{max} \times ((N + N_{1q}) \times N)]$, where $N$ denotes the number of qubits and $N_{1q}$ the number of 1-qubit gates. Each matrix slice within the tensor contains $N$ rows that specify control and target qubit locations in CNOT

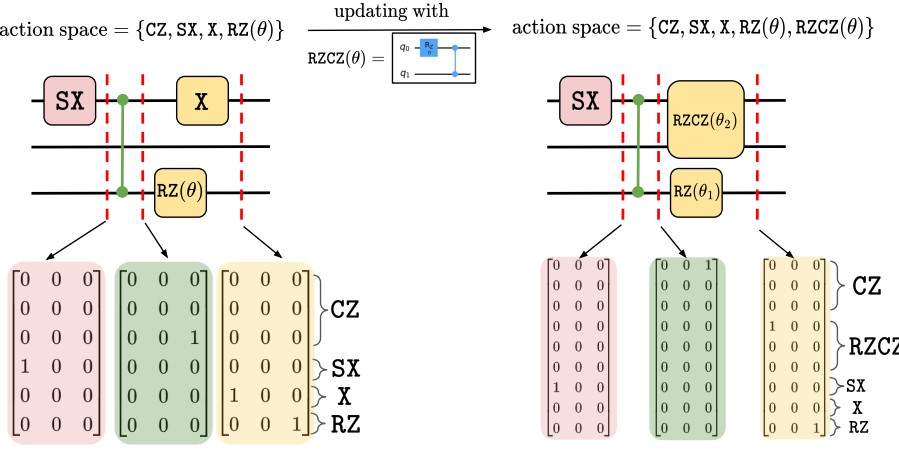

*Figure 6.* **The refined encoding of a parameterized quantum circuits (PQCs) into a tensor**. This is the observable for the reinforcement learning algorithm. The 2-qubit gates are encoded into a matrix whose dimension is dependent on the number of qubits. Meanwhile, the 1-qubit gates are encoded into the remaining $N_{1q}$ rows, which define the number of different 1-qubit gates present in the action space. After the program synthesis algorithm (described in 3.2) finds the most common patterns of gates, i.e. *gadgets*, in the top performing PQCs, the action space is then updated with the extracted *gadget*. In the *gadgetized* reinforcement learning, the dimension of the tensor is then increased. The increase in dimension depends on whether the *gadget* is a 1- or 2-qubit gate.

gates, followed by either 3 rows (for RX, RY and RZ) or 3 rows (for SX, X, RZ) to indicate the positions of 1-qubit gates. When we update the action space by incorporating the *gadgets*, (which are the composite gateset found using the program synthesis algorithm) then, depending on the added gadget, we update the size of the tensor. After *gadgetizing* the action space, we rerun the RL agent with the extended encoding of the PQCs as shown in Fig. 6.

**D.2. Double Deep Q-Network (DDQN)**

Deep Reinforcement Learning (RL) methods employ Neural Networks (NNs) to refine the agent's policy in order to maximize the cumulative return:

$$G_t = \sum_{k=0}^{\infty} \gamma^k r_{t+k+1}, \tag{7}$$

where $\gamma \in [0, 1)$ denotes the discount factor. An action value function is assigned to each state-action pair $(s, a)$, capturing the expected return when action $a$ is taken in state $s$ at time $t$ under policy $\pi$:

$$q_\pi(s, a) = \mathbb{E}_\pi[G_t | s_t = s, a_t = a]. \tag{8}$$

The objective is to find an optimal policy that maximizes the expected return. This can be achieved through the optimal action-value function $q_*$, which satisfies the Bellman optimality equation:

$$q_*(s, a) = \mathbb{E}\left[r_{t+1} + \max_{a'} q_*(s_{t+1}, a') \big| s_t = s, a_t = a\right]. \tag{9}$$

Rather than solving the Bellman equation directly, value-based RL focuses on approximating the optimal action-value function through sampled data. Q-learning, a widely used value-based RL algorithm, initializes with arbitrary Q-values for each $(s, a)$ pair and iteratively updates them to approach $q_*$. The update rule for Q-learning is:

$$Q(s_t, a_t) \quad \leftarrow \quad Q(s_t, a_t) + \alpha\big(r_{t+1} + \gamma \max_{a'} Q(s_{t+1}, a') - Q(s_t, a_t)\big), \tag{10}$$

where $\alpha$ is the learning rate, $r_{t+1}$ is the reward received at step $t + 1$, and $s_{t+1}$ is the resulting state after taking action $a_t$ in state $s_t$. Convergence to the optimal Q-values is guaranteed under the tabular setup if all state-action pairs are visited infinitely often (Melo, 2001). To promote exploration in Q-learning, an $\epsilon$-greedy policy is adopted, defined as:

$$\pi(a|s) := \begin{cases} 1 - \epsilon_t & \text{if } a = \max_{a'} Q(s, a'), \\ \epsilon_t & \text{otherwise.} \end{cases} \tag{11}$$

This $\epsilon$-greedy policy adds randomness during learning, while the policy becomes deterministic after training.

To handle large state and action spaces, NN-based function approximations are used to extend Q-learning. Since NN training relies on independently and identically distributed samples, this requirement is met through experience replay. With experience replay, transitions are stored and randomly sampled in mini-batches, reducing the correlation between samples. For stable training, two NNs are employed: a policy network that is frequently updated, and a target network, which is a delayed copy of the policy network. The target value $Y$ used in updates is given by:

$$Y_{\text{DQN}} = r_{t+1} + \gamma \max_{a'} Q_{\text{target}}(s_{t+1}, a'). \tag{12}$$

In the double DQN (DDQN) approach, the action used for estimating the target is derived from the policy network, minimizing the overestimation bias observed in standard DQN. The target is thus defined as:

$$Y_{\text{DDQN}} = r_{t+1} + \gamma Q_{\text{target}}\big(s_{t+1}, \arg\max_{a'} Q_{\text{policy}}(s_{t+1}, a')\big). \tag{13}$$

This target value is then approximated through a loss function, which in our work is chosen to be the smooth L1-norm given by

$$\text{SmoothL1}(x) = \begin{cases} 0.5x^2 & \text{if } |x| < 1, \\ |x| - 0.5 & \text{otherwise.} \end{cases} \tag{14}$$

### D.3. Reinforcement learning agent hyperparameters

The hyperparameters of the double deep-Q network algorithm were selected through coarse-grain search, and the employed network architecture depicts a feed-forward neural network whose hyperparameters are provided in Tab. 2.

Table 2. GRL and RL agent hyperparameters.

| Parameter | Value |
|---|---|
| Batch size | 1000 |
| Memory size | 20000 |
| Neurons | 1000 |
| Hidden layers | 5 |
| Dropout | 0.0 |
| Network optimizer | Adam (Kingma, 2014) |
| Learning rate | $10^{-4}$ |
| Update target network | 500 |
| Final gamma | $5 \times 10^{-3}$ |
| Epsilon decay | 0.99995 |
| Minimum epsilon | $5 \times 10^{-2}$ |

In the implemented agents, we greedily update the threshold ($\zeta$) after 2000 episodes, with an amortization radius set at $10^{-4}$. This amortization radius decreased by $10^{-5}$ after every 50 successfully solved episode, beginning from an initial threshold value of $\zeta_1 = 5 \times 10^{-3}$. Moreover, in each episode, we set the total number of steps $T_{\text{max}} = 20$ for 2-qubit TFIM and $T_{\text{max}} = 50$ for 3-qubit TFIM.

Throughout this paper, we utilize a gradient-free COBYLA optimizer (Powell, 1994) with hyperparameter settings similar to ref. (Virtanen et al., 2020) and 1000 iterations at each step of an episode to optimize the PQCs.

## E. Ablation study

Table 3 gives a more detailed overview of the results of our experiments.

| Settings | Problem | Field str. | Avg. err. | Avg. gate | Avg. 2q gate | Avg. depth | Min. err. | Min. gate | Min. 2q gate | Min. depth |
|---|---|---|---|---|---|---|---|---|---|---|
| **2-gate GRL** | 2-qubit TFIM | $h=10^{-3}$ | $\mathbf{1.0 \times 10^{-12}}$ | 25.33 | 8.0 | 22.33 | $\mathbf{6.67 \times 10^{-16}}$ | 21 | 3 | 19 |
| | | $h=5\times 10^{-2}$ | $\mathbf{1.5 \times 10^{-11}}$ | 18.0 | 3.67 | 15.0 | $\mathbf{1.8 \times 10^{-12}}$ | 8 | 2 | 7 |
| | | $h=1$ | $3.1 \times 10^{-11}$ | 13.67 | 3.33 | 10.0 | $1.4 \times 10^{-11}$ | 9 | 3 | 7 |
| | 3-qubit TFIM | $h=10^{-3}$ | $\mathbf{6 \times 10^{-9}}$ | 20.5 | 2.5 | 12.0 | $\mathbf{2.6 \times 10^{-9}}$ | 19 | 1 | 12 |
| | | $h=5\times 10^{-2}$ | $1.3 \times 10^{-4}$ | 42.0 | 11.67 | 29.0 | $1.3 \times 10^{-4}$ | 33 | 3 | 19 |
| | | $h=1$ | $\mathbf{0.10}$ | 41.0 | 8.33 | 28.0 | $\mathbf{7.2 \times 10^{-9}}$ | 35 | 5 | 25 |
| **1-gate GRL** | 2-qubit TFIM | $h=10^{-3}$ | $2.1 \times 10^{-10}$ | 18.33 | 5.33 | 14.67 | $3.9 \times 10^{-11}$ | 8 | 2 | 6 |
| | | $h=5\times 10^{-2}$ | $5.3 \times 10^{-10}$ | 14.33 | 3.33 | 11.0 | $2.3 \times 10^{-10}$ | 11 | 2 | 9 |
| | | $h=1$ | $1.5 \times 10^{-10}$ | 11.67 | 1.67 | 8.0 | $6.6 \times 10^{-11}$ | 8 | 1 | 6 |
| | 3-qubit TFIM | $h=10^{-3}$ | $1.2 \times 10^{-7}$ | 43.0 | 11.5 | 28.5 | $1.2 \times 10^{-8}$ | 38 | 6 | 26 |
| | | $h=5\times 10^{-2}$ | $\mathbf{9.6 \times 10^{-4}}$ | 16.0 | 3.67 | 10.33 | $\mathbf{1.3 \times 10^{-4}}$ | 11 | 2 | 6 |
| | | $h=1$ | $0.34$ | 40.67 | 25.67 | 31.0 | $0.26$ | 36 | 19 | 27 |
| **RL only** | 2-qubit TFIM | $h=10^{-3}$ | $6.4 \times 10^{-7}$ | 14.33 | 3.0 | 11.33 | $9.5 \times 10^{-9}$ | 11 | 2 | 9 |
| | | $h=5\times 10^{-2}$ | $1.3 \times 10^{-4}$ | 21.67 | 5.33 | 16.33 | $1.6 \times 10^{-6}$ | 21 | 3 | 15 |
| | | $h=1$ | $5.7 \times 10^{-3}$ | 20.33 | 3.0 | 13.67 | $7.4 \times 10^{-6}$ | 14 | 2 | 10 |
| | 3-qubit TFIM | $h=10^{-3}$ | $7.5 \times 10^{-7}$ | 18.0 | 9.5 | 13.5 | $7.5 \times 10^{-7}$ | 11 | 3 | 6 |
| | | $h=5\times 10^{-2}$ | $1.7 \times 10^{-3}$ | 15.67 | 7.0 | 11.67 | $1.0 \times 10^{-3}$ | 12 | 3 | 9 |
| | | $h=1$ | $0.53$ | 36.0 | 7.0 | 24.3 | $0.39$ | 29 | 2 | 18 |

*Table 3.* Results of the gadget reinforcement learning (GRL) agent on finding the ground state of transverse field Ising model (TFIM) for two and three qubits in three different regimes (low, intermediate and strong transverse field). We compare the performance with one and two extracted gadgets and RL only. The average is taken over different initializations of the neural network and the minimum is the best-performing instance. By looking at the best solution, we see that GRL produces better approximations and sometimes even shorter circuits than RL only, especially in the hardest regimes.

## F. Cumulative rewards and return: 3-qubit TFIM

The Fig. 7 illustrates the cumulative performance of RL and GRL agents over a series of episodes for solving the 3-qubit transverse field Ising model (TFIM). Key metrics, such as error threshold, rewards, and returns, are plotted against the number of episodes to evaluate the effectiveness of each approach.

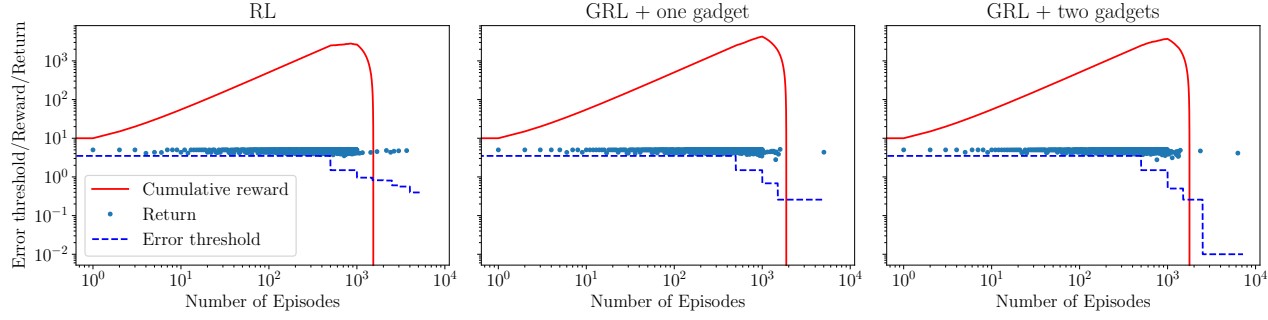

*Figure 7.* **Comparative performance of RL and GRL agents in solving the 3-qubit TFIM**. The plots show cumulative rewards, error scaling, and success rates over episodes for RL-only, GRL with one gadget, and GRL with two gadgets. GRL agents demonstrate improved stability, faster convergence, and higher success rates, particularly when multiple gadgets are incorporated

The RL-only agent struggles to achieve stable rewards across episodes, showing significant fluctuations and slow convergence.

In contrast, GRL agents exhibit steady improvements in cumulative rewards, particularly when gadgets are incorporated into the action space. Error Scaling:

For GRL with one gadget, the error decreases significantly in early episodes but plateaus at a higher value, indicating limited capability to reach machine precision. GRL with two gadgets further reduces the error, demonstrating the benefits of adding more extracted gadgets for addressing complex regimes. Success Rates:

The frequency of successful episodes (defined by achieving the ground state approximation within a predefined threshold) increases with the number of gadgets in the GRL setup. The RL-only agent rarely achieves success, particularly in the harder regimes.

The results highlight the scalability and robustness of GRL agents. Incorporating gadgets not only accelerates the learning process but also enables the agent to achieve lower error thresholds and higher success rates, even for challenging configurations like the 3-qubit TFIM.

This analysis demonstrates the potential of gadget-based reinforcement learning to significantly enhance agent performance. The inclusion of additional gadgets systematically reduces errors and increases the frequency of successful episodes, making this approach a viable solution for tackling more complex quantum systems.

## G. Performance comparison of the transpiled circuits

Here we compare the length and the performance of the circuits obtained to solve the 2 and 3-qubit TFIM ground state at the phase change point ($h = 1$) using the RL agent with a universal gateset (i.e. RX, RY, RZ and CX) and GRL agent with an

| Backend name | GRL | | RL | | Backend name | GRL | | RL | |
|---|---|---|---|---|---|---|---|---|---|
| | Avg. | Min. | Avg. | Min. | | Avg. | Min. | Avg. | Min. |
| fake_torino | **−3.309** | **−3.366** | −3.287 | −3.351 | fake_torino | **−2.188** | **−2.213** | −2.164 | −2.1992 |
| fake_kawasaki | **−3.370** | **−3.397** | −3.235 | −3.319 | fake_kawasaki | **−2.162** | **−2.196** | −2.123 | -2.1592 |
| fake_quebec | **−3.318** | **−3.379** | 3.266 | −3.318 | fake_quebec | **−2.118** | **−2.145** | −2.084 | -2.1597 |

*Table 4.* **Performance comparison of GRL and RL agents for 2-qubit (right table) and 3-qubit (left table) TFIM ground state preparation at the phase transition point** ($h = 1$). The table presents the average and minimum energy values obtained using simulated noisy quantum hardware. GRL, leveraging an extended action space with gates from the IBM Heron processor and an additional gadget, demonstrates consistently better performance compared to RL. This is reflected in lower (more negative) energy values across all backends, highlighting GRL's enhanced optimization capabilities and robustness in circuit design. It should be noted that the true minimum energies for 2- and 3-qubit cases are $−2.236$ and $−3.494$ respectively.

extended action space consisting of gateset of **IBM Heron** and processor and one additional *gadget*. The performance of the GRL and RL are summarized in the Tab. 4. From the table we note the following observations:

1. *Consistent outperformance*: GRL consistently achieves better results than RL across multiple simulated backends. This is evident from the lower (more negative) energy values for GRL in both the 2-qubit and 3-qubit cases.

2. *Improved minimum values*: GRL not only shows better average performance but also achieves lower minimum energy values, indicating its ability to find better solutions more consistently.

3. *Versatility across backends*: The advantage of GRL is maintained across various noisy backends of **IBM Heron** and **IBM Eagle** processors, suggesting its robustness to different quantum hardwares.

4. *Potential for real hardware*: While these results are from simulated noisy environments, they suggest that GRL could offer significant advantages when applied to real quantum hardware, potentially leading to more efficient quantum circuit designs for solving TFIM ground state problems.

GRL's extended action space, which includes the IBM Heron processor's gateset and an additional gadget, likely contributes to its ability to find more optimal circuit configurations.

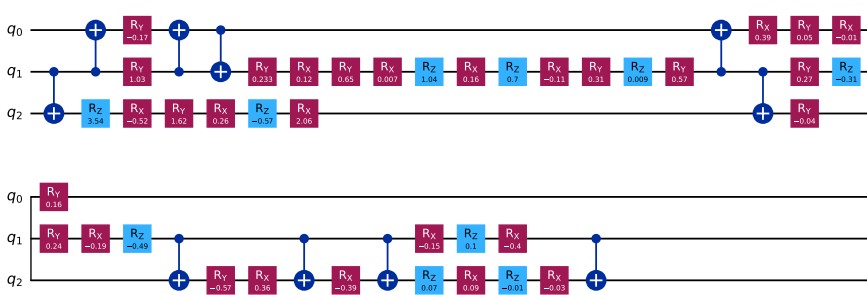

*Figure 8.* **Best-performing circuit obtained from curriculum reinforcement learning (RL) agent in solving $N = 3$ TFIM using a universal gate set**. We train the agent for 5000 episodes and choose the circuit that provides the lowest error in estimating the ground state energy.

In Fig. 8 we illustrate one of the best-performing circuits by the RL. On the other hand, in Fig. 9, we show the best circuit

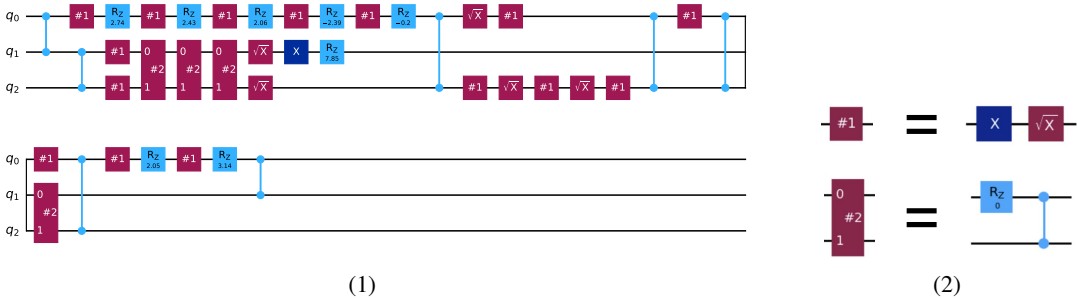

*Figure 9.* (1) **Best-performing circuit obtained from the gadget reinforcement learning (GRL) agent with two gadgets in finding the ground state of a 3-qubit TFIM**. Similar to Fig. 8 we train the GRL agent for 5000 episode and then choose the circuit that gives the lowest error in ground state estimation. In (2) we illustrate the extracted gadgets from easier problems with 2-qubit TFIM.

obtained for solving the same problem using our GRL agent.

Before implementation on real hardware, we would need to transpile the circuits to only use the instructions available on the specific platform. Figure 10 compares the transpiled circuit obtained through the RL agent with a universal gate set with that of our GRL agent with two extracted gadgets. We show a single example as an illustration, please refer to Table 1 in the main text for more quantitative details.

## H. Summary of training time

In Fig. 11 we compare the training time of reinforcement learning (RL) and gadget reinforcement learning (GRL). With a fixed computational budget of 5000 episodes, the GRL agent identified the optimal solution—represented by a parameterized quantum circuit that approximates the TFIM ground state—much faster than the RL-only agent. This demonstrates GRL's ability to achieve the desired accuracy with fewer interactions between the agent and the environment. This advantage makes GRL particularly effective in noisy environments. Moreover, by completing the task in less time, GRL significantly reduces energy consumption and computational resource requirements, making it a practical and efficient solution for resource-constrained scenarios.

## I. IBM Heron processor: IBMQ Torino

Figure 12 shows the topology of the `IBMQ Torino` platform.

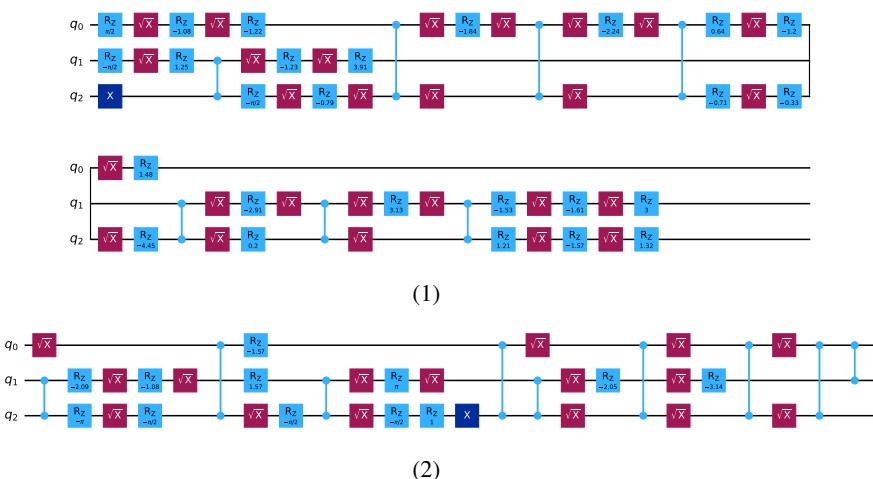

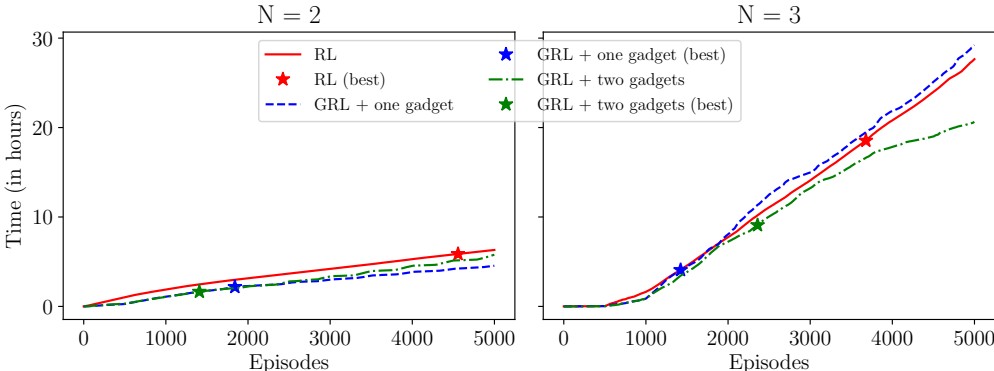

*Figure 10.* **Comparison between the transpiled circuit obtained from (1) reinforcement learning (RL) using a universal gate set, (2) gadget reinforcement learning (GRL) using the native gateset for the `IBM Heron` processor and two gadgets**. After transpilation in real hardware, the circuit produced by GRL is more compact compared to the RL-agent circuit.

*Figure 11.* **Gadget reinforcement learning (GRL) outperforms RL-only agents in finding the optimal solution more quickly for both the $N = 2$ and $N = 3$ qubit transverse field Ising model (TFIM).** The agent was trained with a fixed computational budget, equivalent to 5000 episodes. The GRL agent identifies the optimal solution, represented by a parameterized quantum circuit that generates a state closest to the TFIM ground state, much faster than the RL-only agent. This demonstrates that GRL is resource-efficient and can be executed within a limited time frame.

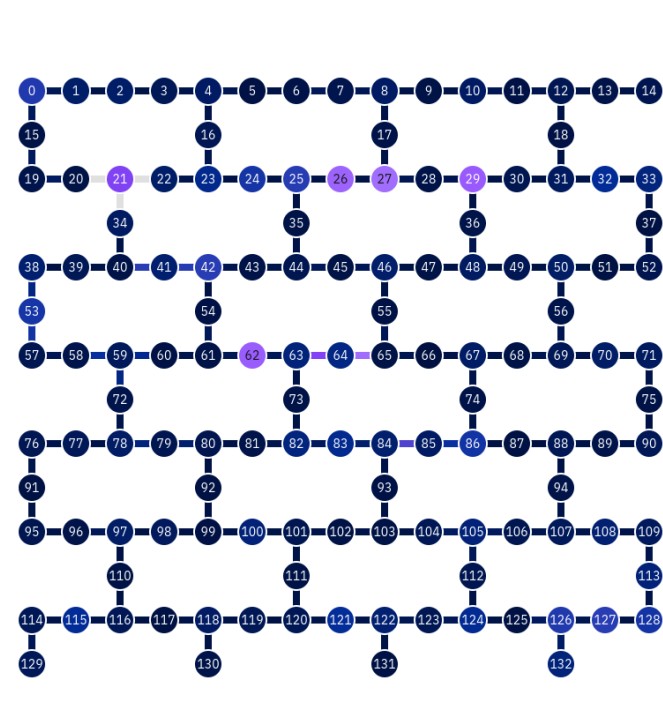

*Figure 12.* The topology of the `IBMQ Torino` which operates on `IBM Heron` processor.

