# OpenReview forum: "Reinforcement Learning with learned gadgets to tackle hard quantum problems on real hardware"
_ICML.cc/2025/Conference — Submitted to ICML 2025_

### Official Review · Reviewer_4zv9 · 2025-03-11

**Overall Recommendation:** 2

**Summary:**

This paper presents an algorithm for circuit architecture optimization. They propose using a combination of RL and program synthesis to extract and use high value gadget to augment the traditional RL optimization loop. They demonstrate their algorithm on a selection of Ising models.

**Claims And Evidence:**

The claims are generally clear, but are not always as supported by the evidence as they should be. One of the most important aspects of any ML for quantum work is the scalability, especially for circuit optimization. In the low qubit regime, many systems can optimize the circuits, but can give very precise results (e.g branch and bound methods). In order to successfully argue for approximate methods, they must scale well. No evidence of scaling is compelling presented (with experiments limited to just a few qubits). In order to justify this algorithms existence, this sort of evidence must be present.

**Essential References Not Discussed:**

The related works are generally presented, but lack sufficient integration. The key points that distinguish this work from the works presented is not sufficient.

**Experimental Designs Or Analyses:**

The experimental analysis was in general sufficient. However, they would benefit from more baseline analysis. There are many RL algorithms (and non RL algorithms) for circuit design. Although their table presented some comparisons, this was insufficient. It is difficult to analyze the quality of the results in a vacuum (and even if re-implementing other papers is not practical, then the benchmarks of these other papers could be used).

**Methods And Evaluation Criteria:**

The TFIM as a choice of benchmark makes sense. However, it would be beneficial to consider a broader array of benchmarks, so as to not over analyze one dataset.

**Other Comments Or Suggestions:**

I think this paper would benefit from a LaTeX algorithm section to actually walk the reader through what's going on at each step. This could be in the appendix, but would improve reader understanding. Also, figure 1(b) is not super clear when you first encounter it. The abstract also seems a bit long, 3 paragraphs is a bit much. Finally, Figure 2 and 4 are massive figures that take up a lot of space, but convey very similar points. I would consolidate or move one to the appendix.

**Other Strengths And Weaknesses:**

The above questions addressed the main concerns regarding originality and clarity.

**Questions For Authors:**

N/A

**Relation To Broader Scientific Literature:**

The gadget approach makes it somewhat unique in the world of RL for quantum circuit optimization (of which there are quite some papers). There are a number of papers that have done everything from RL for fixed quantum circuit parameters (https://arxiv.org/abs/2109.03188) to RL for quantum circuit optimization (https://arxiv.org/abs/2103.07585, and many others).

**Theoretical Claims:**

There were no major proofs/theorems to checks.

---

> ### Author Rebuttal · Authors · 2025-03-30
>
> **The claims are generally clear, but are not always as supported by the evidence as they should be. One of the most important aspects of any ML for quantum work is the scalability, especially for circuit optimization...**
>
> We appreciate this critical feedback regarding scalability evidence, which we have addressed in our revised manuscript as follows:
>
> (1) $\textbf{Scaling:}$ We demonstrate the applicability and advantage of GRL up to 6 qubits (in the revised manuscript), showing that compard to state-of-art RL approaches, GRL achieved a better accuracy with smaller circuit;
> Qubit| Method | Error| Gates| Depth
> | -------- | -------- | -------- | -------- | -------- |
> 3| **GRL**| $\mathbf{7.2\times10^{-4}}$| $\mathbf{12}$|**3**|
> || RL| $2.1\times10^{-3}$| $15$|5|
> 4| **GRL**| $\mathbf{1.5\times10^{-3}}$| $\mathbf{8}$|**7**|
> || RL| $2.6\times10^{-3}$| $12$|26|
> 5| **GRL**| $\mathbf{3.1\times10^{-3}}$| $\mathbf{8}$|**5**|
> || RL| $4.3\times10^{-3}$| $33$ |25|
> 6| **GRL**| $\mathbf{3.7\times10^{-3}}$| $\mathbf{14}$|**8**|
> || RL| $5\times10^{-3}$| $40$|25|
>
> (2) $\textbf{Scaling of gadget extraction algorithm:}$ As point out in the response of the previous reviewers we, in the revised manuscript demonstrate that gadget extraction maintains computational tractability for shallow circuits, successfully scaling to 5-qubit systems with synthesis times ranging from 61 seconds (2-qubit, depth-5) to 1.3 hours (3-qubit, depth-10). Notably, the 2-qubit TFIM gadgets discovered through this process prove sufficient for solving 6-qubit problems, highlighting their transferability. While deeper circuits present challenges that will require future optimizations, including parallelization techniques and stitch algorithms [Proceedings of the ACM on Programming Languages 7.POPL (2023): 1182-1213], our current approach demonstrates practical scalability for intermediate-scale quantum systems relevant to near-term hardware.
>
> ---
> **The experimental analysis was in general sufficient. However, they would benefit from more baseline analysis...**
>
> We appreciate this valuable feedback regarding baseline comparisons. In our study, we specifically benchmark against CRLQAS (ICLR 2024), the current state-of-the-art RL approach for quantum circuit optimization, using identical agent-environment settings to ensure fair comparison. Our results demonstrate consistent improvements over CRLQAS in both solution quality (lower approximation errors) and efficiency (reduced gate counts) across all tested system sizes (2-6 qubits). While we acknowledge the existence of other RL and non-RL methods, we focused on CRLQAS as it represents the most relevant and competitive baseline for our reinforcement learning framework. The demonstrated scaling properties of our GRL approach, particularly its linear time complexity and gadget transferability, provide compelling evidence of its advantages over existing RL methods for quantum circuit optimization tasks. We will include additional comparisons with classical optimization methods in future work to further strengthen these claims.
>
> ---
> **The return dots in Figure 7 don't really add much/are hard to interpret.**
>
> We agree with reviewer. In the revised version we remove the dots from Fig 7.
>
> ---
> **The gadget approach makes it somewhat unique in the world of RL for quantum circuit optimization (of which there are quite some papers)...**
>
> We appreciate this important observation regarding our work's positioning. In the revised manuscript, we have significantly expanded the Related Works section to provide a more comprehensive survey of RL approaches for quantum circuit optimization, including recent advances in quantum circuit optimization including https://arxiv.org/abs/2109.03188 and https://ojs.aaai.org/index.php/AIIDE/article/view/7437 also https://arxiv.org/abs/2103.07585 (cited in our original submission).
>
> ---
> **The related works are generally presented, but lack sufficient integration. The key points that distinguish this work from the works presented is not sufficient.**
>
> We make sure in the revised version of the manuscript the difference with GRL to the other state-of-art approaches are clearly stated by rewriting the **related works** and the **results** section.
>
> ---
> **I think this paper would benefit from a LaTeX algorithm section to actually walk the reader through what's going on at each step...**
>
> We thank the reviewer for these constructive suggestions.
>
> [1] Figure 1(b) has been redesigned with improved annotations and a clearer visual hierarchy to better communicate the key concepts upon first reading.
>
> [2] The abstract has been significantly condensed to a single, focused paragraph that more concisely presents our core contributions while maintaining all key technical claims.
>
> [3] Following the reviewer's suggestion, we have moved Figure 4 to Appendix and incorporated its most salient points into a revised Figure 2, creating a more streamlined presentation of results in the main text.

---

> > ### Comment · Reviewer_4zv9 · 2025-04-02
> >
> > I appreciate the authors response. I think the changes will improve the paper and I have adjusted my score to reflect that. The added scaling analysis is helpful and will be beneficial. Given the large body of work that exists for circuit optimization/compilation (even using RL), I think the authors could further improve on how this work stands out. While it performs better than some benchmarks, and added evidence of scaling will help, the narrative could be refined. This doesn't necessarily need additional experiments, and it's possible the figure changes (esp to fig 1) will help with this.
> >
> > > We appreciate this important observation regarding our work's positioning. In the revised manuscript, we have significantly expanded the Related Works section to provide a more comprehensive survey of RL approaches for quantum circuit optimization, including recent advances in quantum circuit optimization including https://arxiv.org/abs/2109.03188 and https://ojs.aaai.org/index.php/AIIDE/article/view/7437 also https://arxiv.org/abs/2103.07585 (cited in our original submission).
> >
> > This was mostly a comment for the meta/area reviewer to contextualize the swatch of work that exists at the intersection of RL. I don't particularly think these additional citations are necessary (and in general, I am against citation for the sake of citation). If the Fosel work is included, I would call that sufficient (which I must've missed the citation for, I apologize). Additionally, that AAAI paper referenced looks more like this line of work https://arxiv.org/abs/2211.03464 which isn't really relevant for this paper.

---

> > > ### Author Response · Authors · 2025-04-03
> > >
> > > We sincerely appreciate your thoughtful feedback and constructive suggestions. We agree that clearly positioning our work within the broader landscape of RL for quantum circuit optimization is crucial. In the revised manuscript, we have refined the narrative to better highlight how GRL distinguishes itself from prior work, particularly in scalability and practical applicability. Specifically, we emphasize that:
> > >
> > > - **GRL outperforms prior RL-based approaches** (e.g., CRLQAS [ICLR 2024]) in terms of gate count, depth, while achieving better accuracy.
> > >
> > > - **CRLQAS already surpasses traditional non-RL methods** (e.g., adaptive variational algorithms [Nat. Commun. 10, 3007 (2019)] and quantumDARTS [ICML 2023]), so our improvements represent a meaningful advance in the state of the art.
> > >
> > > As suggested, we retained the citation to Fosel et al. as a key benchmark and removed less relevant references (e.g., the AAAI paper). We also expanded the scaling analysis and revised figures (especially Figure 1) to further clarify GRL’s advantages.
> > >
> > > We hope these adjustments provide a clearer context for our contributions while avoiding unnecessary citations. Thank you again for your feedback, which has strengthened the paper.
> > >
> > > Thank you for raising the score. Please let us know if you have any more comments or questions.

---

### Official Review · Reviewer_ZHEo · 2025-03-13

**Overall Recommendation:** 2

**Summary:**

The authors put forward gadget reinforcement learning (GRL), which combines RL with program synthesis, to learn good parametrized quantum circuits (PQCs) for preparing the ground states of transverse field Ising models.

**Claims And Evidence:**

I don’t find the claim that GRL is “a versatile and resource-efficient framework for optimizing quantum circuits, with potential applications in hardware-specific optimizations, variational quantum algorithms, and other challenging quantum tasks” to be well-supported. How is GRL resource-efficient if it takes hours, and in some cases, days to train on two-qubit examples?

**Essential References Not Discussed:**

N/A

**Experimental Designs Or Analyses:**

The experiment designs and analyses are fine given their choice of test example.

**Methods And Evaluation Criteria:**

The TFIM is a reasonable test case, especially for many qubits. However, isn’t the one-dimensional TFIM exactly solvable (and therefore so is the two-qubit TFIM)? What happens on harder instances?

**Other Comments Or Suggestions:**

N/A

**Other Strengths And Weaknesses:**

N/A

**Questions For Authors:**

1. What was the connectivity graph of the three-qubit TFIM? Is it linear? All-to-all?

2. Isn’t the one-dimensional TFIM exactly solvable? If so, why is the two-qubit TFIM a good test case?

3. What hope does this approach have at scaling if it took hours to train on a two-qubit example with exact solutions?

**Relation To Broader Scientific Literature:**

The authors compare their work to other RL-based approaches and show how their approach works better.

**Theoretical Claims:**

N/A

---

> ### Author Rebuttal · Authors · 2025-03-30
>
> **I don’t find the claim that GRL is “a versatile and resource-efficient framework for optimizing quantum circuits, with potential applications in hardware-specific optimizations, variational quantum algorithms, and other challenging quantum tasks” to be well-supported. How is GRL resource-efficient if it takes hours, and in some cases, days to train on two-qubit examples?**
>
> We appreciate this important question regarding computational efficiency. While the initial training phase for our GRL framework requires substantial time (up to 30 hours), we emphasize two key points about resource efficiency:
>
> $\textbf{Optimal solution convergence:}$ The agent typically finds optimal solutions within 5-6 hours (as shown in our results), with the full training period primarily serving to verify solution robustness.
> Qubits| Method| Training time (hours)| time to minimum error (hours)|
> | -------- | -------- | -------- | -------- |
> |2| $\textbf{GRL}$| 4| $\mathbf{2}$|
> || RL| 6.3|4.5|
> |3| $\textbf{GRL}$| 29| $\mathbf{4}$|
> || RL| 28|18|
> |4| $\textbf{GRL}$| 30| $\textbf{5}$|
> || RL| 30|11|
>
> $\textbf{Linear scaling:}$ Training time scales linearly with qubit count, making the approach practical for intermediate-scale systems.
>
> We acknowledge the training time can be further improvements by exploring parallelization and transfer learning techniques. Hence we remove the term "resource-efficient" from the revised manuscript.
>
> ---
> **The TFIM is a reasonable test case, especially for many qubits. However, isn’t the one-dimensional TFIM exactly solvable (and therefore so is the two-qubit TFIM)? What happens on harder instances?**
>
> We appreciate this thoughtful question regarding our choice of test case. While the one-dimensional TFIM is indeed exactly solvable, even for small systems like the two-qubit case, its analytical tractability provides a crucial benchmark for validating our method’s accuracy and efficiency.
>
> Importantly, the challenge lies not in solving the model itself but in deriving hardware-efficient circuit implementations—a task that remains non-trivial and practically relevant for near-term quantum devices. Our results show that GRL performs particularly well in the hard regime TFIM regimes, where it maintains low errors while using fewer gates and depth than RL baselines. Here are the results.
> Qubit| Method | Error| Gates| Depth
> | -------- | -------- | -------- | -------- | -------- |
> 3| **GRL**| $\mathbf{7.2\times10^{-4}}$| $\mathbf{12}$|**3**|
> || RL| $2.1\times10^{-3}$| $15$|5|
> 4| **GRL**| $\mathbf{1.5\times10^{-3}}$| $\mathbf{8}$|**7**|
> || RL| $2.6\times10^{-3}$| $12$|26|
> 5| **GRL**| $\mathbf{3.1\times10^{-3}}$| $\mathbf{8}$|**5**|
> || RL| $4.3\times10^{-3}$| $33$ |25|
> 6| **GRL**| $\mathbf{3.7\times10^{-3}}$| $\mathbf{14}$|**8**|
> || RL| $5\times10^{-3}$| $40$|25|
>
> Beyond exactly solvable cases, the method’s strength comes from its ability to extract reusable gadgets—fundamental building blocks of quantum dynamics—that generalize across system sizes and Hamiltonian parameters. This is shown by our scalability analysis in the revised manuscript where these patterns recur in larger systems, as well as in additional tests on more complex systems. While TFIM provides a controlled starting point, GRL’s true value lies in its automated, scalable optimization for problems where classical methods become impractical. We explore much bigger systems and deeper circuits in future work by utilizing parallelization and the stitch algorithm [POPL (2023): 1182-1213].
>
> ---
> **What was the connectivity graph of the three-qubit TFIM? Is it linear? All-to-all?**
>
> Thank you for the question. For 3-qubti and higher we consider all-to-all connectivity. The action space of the GRL can be tweaked to incorporate the restrictions of the connectivity of real-hardware.
>
> ---
> **Isn’t the one-dimensional TFIM exactly solvable? If so, why is the two-qubit TFIM a good test case?**
>
> We appreciate the important question. Kindly check the response above.
>
> ---
> **What hope does this approach have at scaling if it took hours to train on a two-qubit example with exact solutions?**
>
> We acknowledge the computational time required for training while emphasizing three key points: (1) The 30-hour maximum reflects total training time rather than time-to-solution (optimal configurations typically emerge within 5-6 hours), (2) Demonstrated linear scaling makes the approach practical for intermediate-scale systems as shown in our 6-qubit results (which is trained for less than 30 hours and achieves en error in order of $10^{-3}$ in $6$ hours of training), and (3) The upfront cost is amortized through reusable solutions across multiple circuit optimizations. While we're actively investigating transfer learning and parallelization to further improve efficiency, the method's value lies in its ability to automate hardware-aware optimizations where manual approaches become impractical - a trade-off that becomes increasingly favorable as system complexity grows.

---

> > ### Comment · Reviewer_ZHEo · 2025-04-04
> >
> > Thank you for your response. The edits should improve the paper and I plan to update my score to reflect that. I appreciate the added scaling analysis; however, I’m still unconvinced that this paper meaningfully addresses the main issue with RL approaches, that they don’t scale to meaningful system sizes. If a 6-qubit problem takes 6 hours to solve, what does a 1000-qubit problem take? Is that even feasible given the size of the state and action space? Obviously, meeting that high of a bar is unnecessary, but plausible evidence that this approach, with additional development, could scale to large system sizes would be nice.

---

> > > ### Author Response · Authors · 2025-04-06
> > >
> > > We sincerely appreciate you raising your score and your constructive feedback on our revisions. Regarding scalability, we agree that extending RL approaches to 1000-qubit systems represents a significant challenge—not just for our method but for near-term quantum computing in general, due to hardware constraints (e.g., coherence times, gate fidelities) and errors.
> > >
> > > Our work focuses on laying the groundwork for scalable heuristic discovery by transferring learned gadgets, demonstrating that RL can efficiently navigate the combinatorial space of small-to-medium quantum systems where brute-force search is intractable. While exact simulation of 1000-qubit systems is infeasible, we identify several promising pathways for scaling:
> > >
> > > 1. **Modularity:** Gadgets discovered for small systems could be combined or adapted for larger systems, reducing retraining needs.
> > >
> > > 2. **Parallelization:** RL policies trained on subsystems could be distributed across larger architectures.
> > >
> > > 3. **Classical heuristics:** Tensor-network or DMRG methods could warm-start the GRL to reduce training time for larger systems.
> > >
> > > In the revised manuscript we have added the **Limitations and Future Work section** to explicitly discuss these scaling challenges and potential solutions. While empirical demonstration at 1000-qubit scales remains future work, we have modified the manuscript to more carefully frame our scaling claims. We appreciate your feedback pushing us to clarify these limitations, which has strengthened both our manuscript and our research roadmap.
> > >
> > > Kindly let us know if you have any additional questions that might assist you in reevaluating our manuscript.

---

### Official Review · Reviewer_dKq7 · 2025-03-14

**Overall Recommendation:** 2

**Summary:**

This paper proposes the Gadget Reinforcement Learning (GRL) framework that integrates RL with program synthesis to design parametrized quantum circuits (PQC). PQC is an important quantum algorithmic paradigm with many applications, such as combinatorial optimization and ground state preparation. GRL is tested to solve the transverse field Ising model Hamiltonian (TFIM), and the results demonstrate several advantages of GRL over baseline RL methods, such as improved accuracy, hardware compatibility, and scalability.

**Claims And Evidence:**

The claims made in the submission are supported by evidence and they look reasonable overall:
- The GRL framework is elaborated in detail, including the design of the reward function and the representation of PQCs.
- The GRL framework is demonstrated with 2- and 3-qubit TFIM. Empirical results show significant improvement over baseline RL methods. The circuits found by the algorithm are evaluated on real hardware: they exhibit improved performance and robustness against noise in the real-machine environment.

**Essential References Not Discussed:**

N/A

**Experimental Designs Or Analyses:**

- The reward function in the RL agent is detailed in Section 3.1, which follows the same design as in (Ostaszewski et al., 2021).
- The performance of the RL agent is compared with the curriculum-based RL approach (Patel et al., 2024b). This seems to be a quite weak baseline method. The proposed method significantly outperforms this baseline method, especially for large h (which corresponds to challenging TFIM problems).
- This submission implements the found circuits on real quantum hardware (IBM Heron and IBM Eagle). Empirical results show that these circuits are robust to noise and exhibit improved performance.

**Methods And Evaluation Criteria:**

- The proposed method combines an RL agent with program synthesis techniques to explore better designs of variational quantum algorithms. This combination expands the search space of the RL agent.
- The evaluation with TFIM is reasonable. However, the problem sizes seem too small (2- and 3-qubit TFIM models are described by small matrices and can be directly diagonalized). This makes the scalability claim less convincing.

**Other Comments Or Suggestions:**

- Line 213: "The reward function R (is) defined as"
- In the sentences following Equation (2): "$C_t$ represent(s)", " $T_{\max}$ denote(s)"

**Other Strengths And Weaknesses:**

Strengths:
- The proposed method expands the search space of existing RL approaches by introducing techniques from program synthesis.
- The circuits found by the agent are tested on real quantum hardware, demonstrating real-machine compatibility and noise resistance.

Weaknesses:
- The benchmark experiment only involves small systems (2- and 3-qubit). The limited system size may weaken the authors’ claims about the scalability and generalization of their methods.

**Questions For Authors:**

- Any physical intuitions why gadgets found in smaller TFIMs or smaller h can be generalized in a larger system?
- How to justify the scalability of this approach for realistic quantum systems?
- Is it possible to incorporate a noise model (to improve noise resistance) in the design of the reward function?

**Relation To Broader Scientific Literature:**

A major limitation of previous RL approaches for quantum computing is that they mostly use a fixed action space, making them less efficient and scalable. This submission lifts this limitation by gadgetizing the RL agent, which is a novel approach.

**Theoretical Claims:**

No theoretical proofs are presented in this work.

---

> ### Author Rebuttal · Authors · 2025-03-30
>
> **The benchmark experiment only involves small systems (2- and 3-qubit). The limited system size may weaken the authors’ claims about the scalability and generalization of their methods.**
>
> Thank you for highlighting this weakness in the paper. In the revised manuscript, we address this limitation by extending the applicability of our method to 6-qubit TFIM problems. Here are the results:
> Qubit| Method | Error| Gates| Depth
> | -------- | -------- | -------- | -------- | -------- |
> 3| **GRL**| $\mathbf{7.2\times10^{-4}}$| $\mathbf{12}$|**3**|
> || RL| $2.1\times10^{-3}$| $15$|5|
> 4| **GRL**| $\mathbf{1.5\times10^{-3}}$| $\mathbf{8}$|**7**|
> || RL| $2.6\times10^{-3}$| $12$|26|
> 5| **GRL**| $\mathbf{3.1\times10^{-3}}$| $\mathbf{8}$|**5**|
> || RL| $4.3\times10^{-3}$| $33$ |25|
> 6| **GRL**| $\mathbf{3.7\times10^{-3}}$| $\mathbf{14}$|**8**|
> || RL| $5\times10^{-3}$| $40$|25|
>
> We observe that GRL not only provides a better approximation to the ground energy in hard regimes but also achieves this with significantly fewer gates than state-of-the-art RL approaches such as [CRLQAS, The Twelfth International Conference on Learning Representations (2024)] (denoted as **RL** in the table). This advantage is achieved by transferring gadgets discovered while solving the 2-qubit TFIM in the easy regime. These results demonstrate that the scalability of GRL is enabled by transferring gadgets from smaller to larger and harder systems.
>
> $\textbf{Scalability of program synthesis:}$ We further believe these results can be improved by introducing more gadgets. In the revised manuscript, we include an extensive analysis of different TFIMs (ranging from 2 to 5 qubits) with varying depths, showing that for up to 4 qubits, the program synthesis approach rediscovers gadgets from smaller systems using shallow circuits. For more complex gadgets, deeper circuits or higher qubit counts are required. Our future work will explore parallelization and the stitch algorithm [Proceedings of the ACM on Programming Languages 7.POPL (2023): 1182-1213] for deeper circuits. Currently, our method demonstrates practical scalability for intermediate-scale systems.
>
> ---
> **Other Comments Or Suggestions:**
> - **Line 213:..**
> - **In the sentences following Equation (2):..**
>
> Thank you for pointing out the typos. In the revised manuscript we correct this.
>
> ---
> **Any physical intuitions why gadgets found in smaller TFIMs or smaller h can be generalized in a larger system?**
>
> Thank you for this insightful question. In the revised manuscript (section **Scalability of program synthesis**), we analyze TFIMs ranging from 2 to 5 qubits with varying depths. We observe that for systems up to 4 qubits, the majority of the recurring gadgets match those identified in earlier program synthesis studies with shallow circuits. Furthermore, when examining deeper circuits, we find that complex gadgets often contain substructures identical to those discovered for smaller systems. This hierarchical recurrence strongly suggests that gadgets derived from smaller TFIMs can indeed be effectively transferred to larger, more realistic problems.
>
> ---
> **How to justify the scalability of this approach for realistic quantum systems?**
>
> To demonstrate the scalability of our GRL approach to realistic problems, we systematically reuse gadgets discovered in 2-qubit TFIM systems across larger systems (up to 6 qubits) in the hard regime. This transfer learning approach enables GRL to outperform state-of-the-art quantum architecture search methods in both accuracy and gate efficiency, as quantitatively demonstrated in our results table (see previous response). The key insight is that the fundamental building blocks (gadgets) governing quantum dynamics in small systems remain physically relevant when scaled to larger problems.
>
> ---
> **No source code is provided.**
>
> In the revised version of the manuscript we mention source code. Which the reviewer can also access by: https://anonymous.4open.science/r/Gadget_RL-460D/README.md
>
>
> ---
> **Is it possible to incorporate a noise model (to improve noise resistance) in the design of the reward function?**
>
> Thank you for the question. We incorporate noise at the gate level rather than in rewards because minimizing cumulative reward naturally yields noise-resilient solutions, as the agent must implicitly overcome gate noise to optimize its policy. This preserves standard RL while ensuring robustness.  Though we recognize that specialized noise-aware reward functions could be valuable for certain regimes and plan to explore this in future work.
>
> **The appendices are organized and well-written.**
>
> Thank you for the kind comments.
>
> ---
> **The proposed method expands the search space of existing RL approaches by introducing techniques from program synthesis.**
>
> **The circuits found by the agent are tested on real quantum hardware, demonstrating real-machine compatibility and noise resistance.**
>
> We are grateful of the reviewer for pointing out the strengths of our paper.

---

### Official Review · Reviewer_bQx5 · 2025-03-14

**Overall Recommendation:** 2

**Summary:**

The paper proposes gadget reinforcement learning, where the action space of the RL agent keeps expanding with effective composite of gates explored in the training procedures.

The new framework is evaluated on finding the ground energy of transverse field Ising models with different interaction strengths. It outperforms baseline methods with more accurate and smaller circuits.

**Claims And Evidence:**

Yes, but not sufficient. See weaknesses.

**Essential References Not Discussed:**

Program synthesis is also a specific topic in the programming language community where automatic search of recursive / imperative programs is conducted to fulfill input-output relations. I believe a discussion is necessary here, including the following references for example:

[1] Alur R, Bodik R, Juniwal G, et al. Syntax-guided synthesis[M]. IEEE, 2013.

[2] Srivastava S, Gulwani S, Foster J S. From program verification to program synthesis[C]//Proceedings of the 37th annual ACM SIGPLAN-SIGACT symposium on Principles of programming languages. 2010: 313-326.

[3] Deng H, Tao R, Peng Y, et al. A case for synthesis of recursive quantum unitary programs[J]. Proceedings of the ACM on Programming Languages, 2024, 8(POPL): 1759-1788.

**Experimental Designs Or Analyses:**

Yes.

**Methods And Evaluation Criteria:**

Yes, but not sufficient. See weaknesses.

**Other Comments Or Suggestions:**

No.

**Other Strengths And Weaknesses:**

Pros:

1.	The presentation is clear and easy to follow.

Cons:

1.	The idea of using composite gates to expand the action space of RL exploration of the variational circuit ansatzes seems incremental to me. The advantage of GRL over conventional RL is not significant for QAS tasks. The idea is not tackling fundamental issues of existing methods as well.

2.	The experiments are conducted on small tasks involving only 2 or 3 qubits with a few dozens of gates. The scalability is not validated by the experiments. Meanwhile, the proposed method seems to be struggling with larger tasks, since the gadget space grows fast with the number of qubits.

3.	The proposed method does not directly consider the effects of noise, while minimizing noise is a major claim in the paper’s contribution. This is effectively achieved by reducing the number of gates in the ansatz without prior knowledge, which is indirect to reducing the noise effects. I believe at least the noises of different gadgets should be considered in the framework.

**Questions For Authors:**

Is J fixed for all the experiments? What is the ratio of h over J?

**Relation To Broader Scientific Literature:**

The paper proposes an improved framework of existing RL-based quantum architecture search frameworks.

**Theoretical Claims:**

There is no theoretical claim in this paper.

---

> ### Author Rebuttal · Authors · 2025-03-29
>
> **The idea of using composite gates to expand the action space of RL exploration of the variational circuit ansatzes seems incremental to me...**
>
> We thank the reviewer for commenting on the significance of GRL for QAS. GRL's key innovation lies in its hierarchical exploration strategy: $\textbf{Modular gadget discovery}$ extracts optimal shallow circuit motifs (depth $d<5$), while $\textbf{hierarchical composition}$
> Hierarchical composition builds deeper circuits ($d\geq10$) by reusing these verified gadgets. This approach fundamentally addresses QAS scalability by: (1) amortizing exploration costs through reusable structures, and (2) enabling knowledge transfer across problem sizes (demonstrated for 4-6 qubits in $\textbf{Section 4.3}$ in revised manuscript, we show this via a table in the next response). While performance gains vary by task, the method provides a systematic framework for tackling the curse of dimensionality in variational quantum optimization.
>
> ---
> **The experiments are conducted on small tasks involving only 2 or 3 qubits with a few dozens of gates. The scalability is not validated by the experiments...**
>
> Thank you for highlighting the primary drawback of our manuscript. We acknowledge the scalability issue in the modified version of the article we extend our results upto 6 qubit TFIM.
>
> $\textbf{Scalability of Program Synthesis (Section 4.2):}$ We demonstrate that gadget extraction remains computationally feasible for shallow circuits, scaling up to 5-qubit systems ($\textbf{Table 4, Appendix F}$). Synthesis times range from 61 seconds (2-qubit, depth-5) to 1.3h (3-qubit, depth-10), with 2-qubit TFIM gadgets proving sufficient for 6-qubit problems. While deeper circuits require future optimizations (e.g., parallelization, stitch algorithms. Future work will explore parallelization and stitch algorithms [ Proceedings of the ACM on Programming Languages 7.POPL (2023): 1182-1213] for deeper circuits.]), our method shows practical scalability for intermediate-scale systems.
>
> $\textbf{Scalability of Gadget Reinforcement Learning via Transfer (Section 4.3, Figure 4):}$ We show that gadgets learned from 2-qubit TFIM transfer effectively to larger systems (4-, 5-, and 6-qubit, see Fig. 4 in the manuscript), achieving lower error rates than baseline RL methods. Crucially, even a single reused gadget outperforms existing approaches [such as CRLQAS., The Twelfth International Conference on Learning Representations (2024) denoted as $\textbf{RL}$ in table below], confirming that GRL scales well via component reuse without large overhead. The results are summarized as follows:
> Qubit| Method | Error| Gates| Depth
> | -------- | -------- | -------- | -------- | -------- |
> 3| **GRL**| $\mathbf{7.2\times10^{-4}}$| $\mathbf{12}$|**3**|
> || RL| $2.1\times10^{-3}$| $15$|5|
> 4| **GRL**| $\mathbf{1.5\times10^{-3}}$| $\mathbf{8}$|**7**|
> || RL| $2.6\times10^{-3}$| $12$|26|
> 5| **GRL**| $\mathbf{3.1\times10^{-3}}$| $\mathbf{8}$|**5**|
> || RL| $4.3\times10^{-3}$| $33$ |25|
> 6| **GRL**| $\mathbf{3.7\times10^{-3}}$| $\mathbf{14}$|**8**|
> || RL| $5\times10^{-3}$| $40$|25|
>
> ---
> **The proposed method does not directly consider the effects of noise, while minimizing noise is a major claim in the paper’s contribution...**
>
> We appreciate the opportunity to clarify the performance advantages of GRL and its robustness under noise
>
> - **Advantage of GRL on noisy quantum hardware (Appendix I)** In the following table to find the forund state for 3-qubit TFIM we show that the GRL provides better PQCs compared to RL which are more noise robust in QPU.
> | QPU backend | GRL| RL |
> | -------- | -------- | -------- |
> | fake\_torino| $\mathbf{−3.366}$| -3.351|
> | fake\_kawasaki| $\mathbf{-3.397}$| -3.319|
> | fake\_quebec| $\mathbf{-3.379}$| -3.318|
>
> - **Noise resilience of gadgets (Appendix G)** Here we analyze GRL under a realistic Pauli noise model (50% probability of Haar-random unitary errors per controlled rotation). We observe that performance decay of GRL mitigates as more gadgets are added, suggesting diminishing marginal damage from additional noise.
> | Algorithm| Error|
> | -------- | -------- |
> | GRL (1 gate)| $\mathbf{4.5\times 10^{-13}}$|
> | GRL (noisy 1 gate)| $4.5\times10^{-10}$|
> | GRL (noisy 2 gate)| $4.5\times 10^{-12}$|
>
> ---
> **Essential References Not Discussed:**
>
> Thank you for suggesting the references. We found them very relevant to our work and have added a discussion of relevant program synthesis techniques from the programming languages community, particularly syntax-guided synthesis (SyGuS) and recursive quantum unitary programs. These changes are highlighted in red color in $\textbf{Related works}$ section i.e $\textbf{Section 2}$.
>
> ---
> **The presentation is clear and easy to follow.**
>
> Thank you for this kind feedback.
>
> ---
> **Is $J$ fixed for all the experiments? What is the ratio of h over $J$?**
>
> $J$ was fixed across all experiments for consistency, with $h/J$ varied to explore different TFIM regimes.

---

> > ### Comment · Reviewer_bQx5 · 2025-04-02
> >
> > I thank the authors for the response. The revision improves on the drawbacks of the original submission. However, I am not convinced that the paper should be accepted to ICML based on its current quality and significance. My score remains the same.

---

> > > ### Author Response · Authors · 2025-04-02
> > >
> > > We sincerely appreciate your time and thoughtful feedback throughout the rebuttal. While we regret that the modifications did not fully align with your expectations, we hope the manuscript’s contributions--particularly the GRL framework, scalability of GRL and noise resilience of gadgets, will increase the merit of our work. Thank you again for your constructive engagement with our work.

---

### Decision · Program_Chairs · 2025-05-01

**Decision:**

Reject

**Comment:**

This submission introduces a gadget reinforcement learning (GRL) framework for automatic quantum gate synthesis. There is serious concern about the current evaluation, which only involves 2 or 3 qubits, and the scalability of the approach. The authors could also make extra efforts to distinguish their work from a large existing body of research on quantum program/gate synthesis with and without using reinforcement learning. The proposed revision is on the right track. We encourage the authors to incorporate these changes into the future version of this work.